# Listeria motility increases the efficiency of epithelial invasion during intestinal infection

Inge M. N. Wortel[1]☯*, Seonyoung Kim[2]☯, Annie Y. Liu[2], Enid C. Ibarra[2], Mark J. Miller[2]*

1 Data Science, Institute for Computing and Information Sciences, Radboud University, Nijmegen, the Netherlands, 2 Department of Internal Medicine, Division of Infectious Diseases, Washington University School of Medicine, St. Louis, Missouri, United States of America

☯ These authors contributed equally to this work.
* inge.wortel@ru.nl (IMW); mmiller23@wustl.edu (MJM)

**Data Availability Statement:** All code required to reproduce the modelling results in this manuscript as well as the raw data files for the main and supplemental figures are available on Github. The code and data are here: https://github.com/

## Abstract

*Listeria monocytogenes* (Lm) is a food-borne pathogen that causes severe bacterial gastroenteritis, with high rates of hospitalization and mortality. Lm is ubiquitous in soil, water and livestock, and can survive and proliferate at low temperatures. Following oral ingestion of contaminated food, Lm crosses the epithelium through intestinal goblet cells in a mechanism mediated by Lm InlA binding host E-cadherin. Importantly, human infections typically occur with Lm growing at or below room temperature, which is flagellated and motile. Even though many important human bacterial pathogens are flagellated, little is known regarding the effect of Lm motility on invasion and immune evasion.

Here, we used complementary imaging and computer modeling approaches to test the hypothesis that bacterial motility helps Lm locate and engage target cells permissive for invasion. Imaging explanted mouse and human intestine, we showed that Lm grown at room temperature uses motility to scan the epithelial surface and preferentially attach to target cells. Furthermore, we integrated quantitative parameters from our imaging experiments to construct a versatile "layered" cellular Potts model (L-CPM) that simulates host-pathogen dynamics. Simulated data are consistent with the hypothesis that bacterial motility enhances invasion by allowing bacteria to search the epithelial surface for their preferred invasion targets. Indeed, our model consistently predicts that motile bacteria invade twice as efficiently over the first hour of infection.

We also examined how bacterial motility affected interactions with host cellular immunity. In a mouse model of persistent infection, we found that neutrophils migrated to the apical surface of the epithelium 5 hours post infection and interacted with Lm. Yet in contrast to the view that neutrophils "hunt" for bacteria, we found that these interactions were driven by motility of Lm—which moved at least ∼50x faster than neutrophils. Furthermore, our L-CPM predicts that motile bacteria maintain their invasion advantage even in the presence of host phagocytes, with the balance between invasion and phagocytosis governed almost entirely by bacterial motility. In conclusion, our simulations provide insight into host pathogen interaction dynamics at the intestinal epithelial barrier early during infection.

ingewortel/2022-listeria-goblets, and archived here: https://doi.org/10.5281/zenodo.7416374.

**Funding:** M.J.M. was supported by R01-AI077600 funded by the National Institutes of Health-National Institute of Allergy and Infectious Diseases (https://www.niaid.nih.gov/) and I.M.N.W. by RGP0053/2020 through the Human Frontiers Science Program (https://www.hfsp.org/). The funders had no role in study design, data collection and analysis, decision to publish, or preparation of the manuscript.

**Competing interests:** The authors have declared that no competing interests exist.

## Author summary

Many important human bacterial pathogens are motile, yet for many it remains unclear how this motility affects their ability to cause disease. Here, we sought to answer this question for *Listeria monocytogenes* (Lm), a food-borne bacterium that can cause severe gut infections leading to hospitalization or even death.

After being ingested, Lm must engage specific "target cells" in the gut before it can cross the gut lining and cause infection. By imaging Lm interacting with mouse and human intestines, we found that Lm motility facilitates this process: motile Lm could more easily reach and move along the gut lining, allowing them to locate target cells faster than non-motile Lm.

To further understand these dynamics, we built a computer model of Lm gut infection and explored how phagocytes, specifically neutrophils, might interfere with this process. Again, motile bacteria more efficiently located and invaded target cells narrowing the window for neutrophils to capture them. But our simulations also challenge the commonly held view that phagocytes "hunt" bacteria, which move orders of magnitude faster. Instead, phagocytes in our simulations act like "fly paper" to capture bacteria. These findings provide new insights into the early dynamics of bacterial gut infections.

## Introduction

Gastroenteritis and diarrheal diseases are a major source of mortality and morbidity worldwide, especially among children in developing countries [1, 2]. The mucosal barrier of the intestine is exposed continuously to microbes and has evolved a variety of mechanisms to prevent pathogen invasion, including epithelial cell shedding, secretion of antimicrobial peptides and mucus and mucosal antibody production [3–5]. To counter these host defense mechanisms, enteropathogenic bacteria produce virulence factors to promote barrier breach, host cell invasion and inhibit the immune response [3]. Understanding the early stages of bacterial pathogenesis is important for identifying vaccine targets and developing new therapies for bacterial infections.

*Listeria monocytogenes* (Lm) is a food borne pathogen that causes severe life-threatening disease with high hospitalization and mortality rates [6]. Lm is ubiquitous in soil, water and livestock and can survive and proliferate at low temperatures, for example in refrigerated foods such as deli meats, unpasteurized soft cheeses, and smoked fish [7]. Lm expresses a variety of virulence factors at 37 °C [8, 9] that facilitate intracellular invasion and the evasion of host immune responses. One example is the pore forming protein Listeriolysin O that disrupts the phagolysosome membrane, allowing Lm to escape into the cytosol and replicate intracellularly. Another is ActA, which polymerizes host cell actin to propel Lm directly into neighboring cells to infect them [10], evading humoral and innate immunity.

Lm pathogenesis has been widely studied using laboratory mouse models and i.v. infection routes [11]. However, efficient oral infection requires the binding of Lm InlA to E-cadherin, which is human-specific [11, 12]. To overcome this limitation, "murinized" Lm was engineered to express a mutated form of InlA that binds mouse E-cadherin and allows oral infection [13–15]. Alternatively, transgenic mice expressing human E-cadherin can be infected orally with WT (wild-type) Lm [16] to model physiological Lm invasion [17]. Such studies have demonstrated that on the luminal surface of the gut epithelium, E-cadherin is primarily accessible around goblet cells, a subset of secretory epithelial cells. Thus, goblet cells are the preferred target cell for attachment and transcytosis across the epithelial barrier [12, 16, 18,

19]. However, the cell dynamics that lead to these Lm-goblet cell interactions are less well understood.

In contrast to published studies using mouse models, human Lm infections typically occur after the ingestion of contaminated food with bacteria proliferating at or below room temperature (RT, 20–25°C). These are typically flagellated and motile, which is key to colonization and biofilm formation outside the mammalian host [7, 20–23]. Indeed, many clinically important bacterial pathogens are flagellated and motile, including Salmonella, Campylobacter, Helicobacter, Yersinia, Pseudomonas [24–26]. This raises the question: to what degree does bacterial motility determine infection outcomes? While motility was shown to play an important role in the invasion dynamics of Salmonella [27–29], studies with Lm have been contradictory: one study found that Lm flagellar null mutants show similar outcomes in mice and that deletion of the *flaA* gene repressor *mogR* dramatically decreased virulence in vivo [20], while others showed that flagellated Lm has a competitive advantage early during oral infection [22].

Here, we focused our investigations on how Lm motility impacts target cell engagement and invasion efficiency. Two-photon (2P) imaging of explanted mouse and human intestine showed that motile Lm explored the surface of the epithelium and rapidly accumulated around target cells. To predict the consequences of Lm motility for interactions with goblet cells and invasion over time, and under various environmental and bacterial motility scenarios, we constructed a computational model called the cellular Potts model (CPM) [30–32]. Using interacting layers of epithelial cells, motile bacteria, and host phagocytes, our "layered CPM" (L-CPM) integrates quantitative parameters obtained from in vivo, in vitro and explant experiments to simulate host-pathogen dynamics at the epithelium. This model reveals that bacterial motility enhances invasion efficiency by permitting bacteria to rapidly search for and engage permissive target cells. We also investigated whether phagocytes patrolling the apical surface of the epithelium during ongoing infection could limit the invasion of motile bacteria. Our model predicts that bacterial motility remains advantageous for invasion even in the presence of host phagocytes unless phagocytes greatly outnumber target cells. Surprisingly, we predict that this balance between invasion and phagocytosis is governed almost entirely by bacterial motility, with phagocyte motility only important for the phagocytosis of non-motile bacteria. In conclusion, our simulations provide insight into host pathogen interactions and challenge fundamental assumptions regarding how phagocytes might limit bacterial invasion early during infection.

## Materials and methods

### Ethics statement

The use of all laboratory animals was approved and performed in accordance with the Washington University Division of Comparative Medicine guidelines and approved by the Washington University Institutional Animal Care and Use Committee (Animal Welfare Assurance #D16–00245, Protocol No. 19–1016).

Fresh human small intestine resection specimens were obtained through the WUSM Digestive Diseases Research Core Center (Washington University Human Research Protection Office Institutional Review Board: Project Epithelial Responses to Bacteria approval No. 201804112). Formal written consent was obtained for each donor and samples were deidentified to protect their privacy.

### Mouse strains

Mice were bred in house or purchased from JAX labs. B6 mice were bred and housed under specific pathogenfree conditions in the animal facility at the Washington University Medical

Center. The use of all laboratory animals was approved and performed in accordance with the Washington University Division of Comparative Medicine guidelines.

## Bacterial culture

Lm EGD [33], PNF8-GFP [34] and Lm$^{Mu}$ strains [14] were stored as frozen glycerol stocks ($\sim$1x10$^9$/ml) at -80˚C. Bacteria were cultured in BHI medium and harvested during log phase growth for inoculation of tissue and mice [35]. Lm concentrations in culture were estimated from standard growth curves by measuring optical density at 600nm.

## Colony forming unit (CFU) assays

No experiments used death as an endpoint and the Lm challenge doses used are well tolerated in B6 background mice. Mice were infected with 2x10$^7$ Lm$^{Mu}$ by direct luminal injection. Mice were sacrificed 4h.p.i. and the spleens harvested and incubated in $CO_2$-independent media containing 25 µg/ml gentamicin for an hour [36]. Tissues were transferred to a RINO tube containing 0.6ml DPBS and 5 SSB32 stainless steel beads and homogenized, diluted and plated onto BHI agar plates and ChromAgar Listeria selective plates. 1–2 days after plating, colonies were counted and CFU analyzed in R (see "Statistics and group sizes in explant experiments" below). For CFU assays, human and mouse tissues were incubated with gentamicin in DMEM to kill extracellular Lm. Tissues were washed 3x, homogenized and serial dilutions plated on Lm selective media to calculate CFU [36]. In some experiments, the intestine was fractionated into epithelial and LP populations by incubation with EDTA/gentle shaking and cells isolated by flow sorting as previously described [37] to allow CFU measurements for cell subsets.

## In vitro bacterial motility analysis

A 2ml culture of Lm-InlA was grown in 14ml polystyrene round-bottom tube (Falcon Cat No 352057) at RT and 37˚C with shaking at 200rpm to OD600 around 1.2. The sample was diluted 1:10 with BHI and 8 µl of 1:10 dilution was used on a non-charged slide (Globe Scientific Cat No 1324W) to check motility. A 2D time lapse video was recorded for 15 seconds with 250ms time interval (60 time points) and 100ms exposure using an Olympus IX51 inverted microscope with 20X objective and phase dichroic filter. The 2D time lapse video was converted to an Imaris file using Imaris 9.5 (RRID:RRID:SCR_007370) and cells were tracked, and percent motile cells were calculated using a 4 µm track displacement length filter. Tracked cells were imported into *celltrackR* [38] (RRID:SCR_021021), after which mean track speed, mean squared displacement (MSD), and autocovariance/turning angles were computed as described in the package documentation [38] and compared between RT and 37˚C. As a measure of uncertainty in the difference in population mean speeds, the following was repeated 10000 times: individual track speeds from both datasets were pooled and sampled with replacement to obtain a resample of equal size. Within that sample, the difference in population means was then determined to obtain the bootstrap distribution of 10000 estimates and its 99% confidence interval (Fig 1F).

 Lm-RT were then filtered for "motile" tracks, Lm-RTm, and their mean squared displacements were fitted to obtain the motility coefficient M and the persistence time P (see S1 Methods for details).

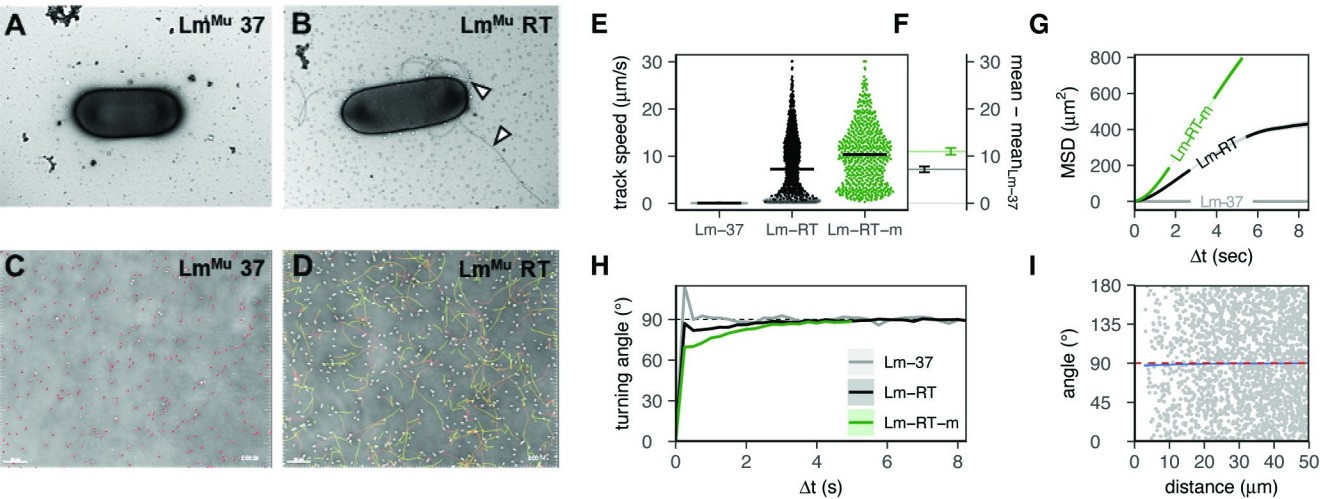

**Fig 1. Murinized Listeria (Lm^Mu) grown at room temperature is flagellated and highly motile.** A: electron microscopy shows that Lm cultured at 37˚C (Lm-37) is non-flagellated, while B: Lm grown at 23˚C (Lm-RT) develops extensive peritrichous flagella (white arrows). Scale bar = 1μm. C: video microscopy shows that Lm-37 is non-motile in vitro (white cells, time encoded tracks) in contrast to D: Lm-RT which is highly motile (white cells, time encoded tracks). Scale bar = 30 μm. This difference in motility is supported by E: the mean speed per bacterium (with Lm-RT-m a filtered, "motile" subset; see S2 Fig), and F: the bootstrapped 99% confidence interval (CI) of the increase in population mean speed for Lm-RT(-m) versus Lm-37 (bootstrap N = 10000). The increased motility of Lm-RT(-m) also yields higher G: mean squared displacements (MSD±standard error, SE) for time intervals Δt, and lower H: turning angles (mean ±SE); mean angles <90˚ indicate persistent motion over interval Δt. Lm-RT-m cells move independently from each other, as shown by I: angles versus distance of single "steps" co-occurring at the same time (gray points reflect 2% of data to avoid overplotting). The mean angle (blue line) is 90˚ if bacteria move independently, with deviations indicating crowding effects (if limited to small distances) or global directionality (if systematic).

### In vitro blood neutrophil and bacterial motility analysis

LysM-GFP mice were sacrificed and immediately $\sim$100 μl of blood was harvested by cardiac puncture and placed in a microfuge tube with 100 μl of $CO_2$ independent media (Gibco) and mixed to delay clotting. Diluted blood $\sim$100 μl was placed on uncharged silane treated glass microscope slides and incubated at RT for 30–60 min until blood had clotted and sera pooled on top. Clots and sera were gently lifted off the slide by pipetting leaving some clotted material adhered to the slide and approximately 30 μl. 1x10^8 Lm-RT (BacLight-Red) was pipetted onto the slide and mounted with a full-length silane treated cover slide. Slides were placed on a pre warmed stage ($\sim$35–37˚C) and then imaged with 2P microscopy with frame rate of 16s for capturing neutrophil behaviors and 100msec for tracking and analyzing Lm motility.

### Electron microscopy of bacterial flagella

Murinized Listeria Monocytogenes (Lm-InlA) were grown in BHI media at RT and 37˚C overnight with shaking at 200rpm. OD600 was measured and overnight culture was diluted to OD600 = 0.6 with BHI. Motility was confirmed by taking a 2D time lapse video recorded for 15 seconds with 250ms time interval (30 time points) and 100ms exposure using an Olympus IX51 inverted microscope with 20X objective and phase dichroic filter. Lm-InlA grown at RT was motile while Lm-InlA grown at 37˚C was non motile. OD600 = 0.6 culture bacteria were fixed with 1% glutaraldehyde (Ted Pella Inc., Redding CA) and allowed to absorb onto freshly glow discharged formvar/carbon-coated copper grids for 10 min. Grids were then washed in dH2O and stained with 1% aqueous uranyl acetate (Ted Pella Inc.) for 1 min. Excess liquid was gently wicked off and grids were allowed to air dry. Samples were viewed on a JEOL 1200EX transmission electron microscope (JEOL USA, Peabody, MA) equipped with an AMT 8-megapixel digital camera (Advanced Microscopy Techniques, Woburn, MA).

## Lm transit time and temperature dependent motility in mouse gut

C57BL/6 mice were infected orally with $2x10^8$ EGD-GFP by gavage using a soft plastic gavage needle to minimize damage to the esophagus. Our rationale for using EGD-GFP is that because EGD binds mouse E-cadherin poorly, it allows us to assess in vivo temperature-dependent changes in motility itself, independent of epithelial binding. Mice were sacrificed 1–2hpi and the ileum harvested, secured to plastic cover slip and the luminal surface exposed for 2P microscopy. Image recordings (100ms time resolution) were acquired near the surface of the epithelium and in the fluid phase above to determine if Lm had entered the ileum and retained motility. Rare cells that moved horizontally through the imaging plane were tracked (Imaris) to demonstrate motility and epithelial scanning.

## 2P imaging of neutrophil recruitment in vivo and ex vivo

To quantify the statistics of neutrophil migration on the epithelium for model parametrisation, LysM-GFP reporter mice were anesthetized, and an incision made in the lower abdomen to expose the ileum for intraluminal infection, which better synchronizes Lm invasion events for 2P imaging. Mice were placed in a warmed imaging stage and a region of the ileum was secured to a plastic coverslip support for 2P imaging [37]. Mice were given s.c. fluids for experiments lasting more than 2hr. Time-lapse imaging was performed from the luminal surface. Multidimensional datasets were rendered and cells were tracked in Imaris (Bitplane) and motility was assessed using celltrackR/MotilityLab (2Ptrack.net).

To assess the number of neutrophils recruited to the surface of the epithelium, LysM-GFP mice were anesthetized and the ileum surgically exposed. Mice were either sham treated with 200μl of vehicle or infected (intraluminal) with $2x10^8$ Lm. Mice were sacrificed from 3–6hpi and imaged with 2P microscopy. Ileum was explanted and 3D images collected from the luminal side to assess neutrophil (LysM-GFP) recruitment to the surface of the epithelium (31-z steps, 400x400x60μm), multi-dimensional data sets were rendered, and cell numbers enumerated using the spot function in Imaris (Bitplane).

To image Lm-neutrophil interactions, LysM-GFP mice were infected intraluminal with $1x10^8$ Lm. At 5hpi, mice were sacrificed, ileum explanted, rechallenged with $1x10^8$ Lm (Bac-Light-Red labeled) and imaged with 2P microscopy at either 24s or 100ms time resolutions.

2P imaging was performed with a custom-built dual-laser 2P microscope [39] equipped with a 1.0 NA 20x water dipping objective (Olympus). Samples were excited with a Chameleon Vision II Ti:Sapphire laser (Coherent) tuned from 750–980nm depending on the experiment and fluorescence emission detected by PMTs simultaneously using appropriate emission filters to separate SHG and the various fluorescence signals.

## 2P imaging of bacterial epithelial scanning, attachment, and invasion in explanted mouse intestine

Mice were euthanized and small intestine harvested and placed in $CO_2$ independent media. Intestines were glued to plastic coverslips using VetBond adhesive and gently cut open longitudinally to expose the luminal surface of the epithelium. Explanted tissues were placed in a custom imaging chamber and covered in DMEM without phenol red containing 2μm red fluorospheres (em 625nm) to identify the mucous layer and tetramethyl rhodamine (Rh)-Dextran to assess epithelial integrity and goblet cell secretion. In some experiments, E-cadherin-CFP mice were imaged to assess epithelial cell numbers and dimensions in the intestinal villi [40]. Tissue explants were infected with $1x10^7$ murinized Lm and imaged with 2P microscopy to record Lm scanning behavior as well as epithelial attachment and invasion. Lm attachment

was assessed by examining the epithelium for the presence of Lm and then quantifying attachment by measuring the number of green voxels and applying a conversion factor of 23.6 voxels/Lm (3D).

## 2P imaging of bacterial epithelial scanning, attachment in human intestine

Fresh human small intestine resection specimens ($\sim$4–9 mm$^2$, $\sim$8 samples total) were placed in $CO_2$ independent media at 4˚C for transport to the imaging lab. The muscularis was trimmed away using surgical scissors to prevent intestinal contraction artefacts and the tissue glued to plastic cover slips using VetBond adhesive (3M), soaked in 10kD Rh-dextran and placed in a custom imaging chamber epithelial surface facing up. Epithelial integrity was assessed by verifying that Rh-dextran is excluded from the LP and that epithelial layer continuity is preserved. The epithelium was also assessed using 800nm excitation, which induces a strong intrinsic fluorescence signal (<480nm) in epithelial cells. The epithelium was challenged with 1x10$^7$CFSE-labeled EGD, PNF8-GFP, BacLight-Green labeled 10403 or 10403 flaA deletion mutant Lm [23] or 1.0µm green/yellow carboxylate-Fluorspheres (Thermo-Fisher) in 25 µl of DMEM. The tissue was placed in warm oxygenated DMEM maintained at 37˚C under slow flow (<1ml/min) for 2P time-lapse imaging. Tissues remained viable for 2–4 hours. Lm attachment was assessed by examining the epithelium for the presence of Lm and then quantifying attachment by measuring the number of green pixels and applying a conversion factor of 9.92pixels/Lm.

## Immunofluorescence microscopy (IFM) of tissue sections

After 2P imaging, tissue sections were fixed and analyzed using epifluorescence microscopy to enumerate neutrophils and monocytes in the tissue. The ileum was harvested 4 hpi., fixed in 4% PFA overnight, embedded in OCT compound (Sakura Finetek) and 5–15 µm cryostat sections cut. Sections were stained with antibodies to E-cadherin, Cytokeratin-18 and DAPI. All antibodies and isotype matched control antibodies are commercially available (BioLegend, Invitrogen).

## Statistics and group sizes in explant experiments

In the explant studies shown in Figs 2 and 3, group sizes consist of 3–5 mice or human tissue explants. Measurements were made in a blinded fashion whenever possible. Results were reproduced in a minimum of two independent experiments. Rather than computing p-values, in accordance with more recent recommendations [41] we instead report effect sizes along with their 95% confidence interval (CI) as estimated via bootstrapping. To compare the #Lm attached to goblet cells, we used the bootstrapped fold-change of the means (Lm-RT/Lm-37), which we considered the most interpretable and meaningful effect size estimate. Since the distribution of CFU values (Fig 2H) was extremely skewed, data were plotted on a logarithmic axis and fold-changes were computed on this scale as well (i.e., as the exponent of the difference in means of log-transformed values). Bootstrapping was performed in R (v4.1.3) using R packages *boot* (v1.3.28) and *simpleboot* (v1.1.7), with 10$^5$ bootstrap samples and using the percentile-based CI.

## Cellular Potts model of the epithelium

Our cellular Potts model (CPM) of the epithelium (Figs 4 and 5) was built in Artistoo [42]. An interactive web version is available at: https://ingewortel.github.io/2022-listeria-goblets. This

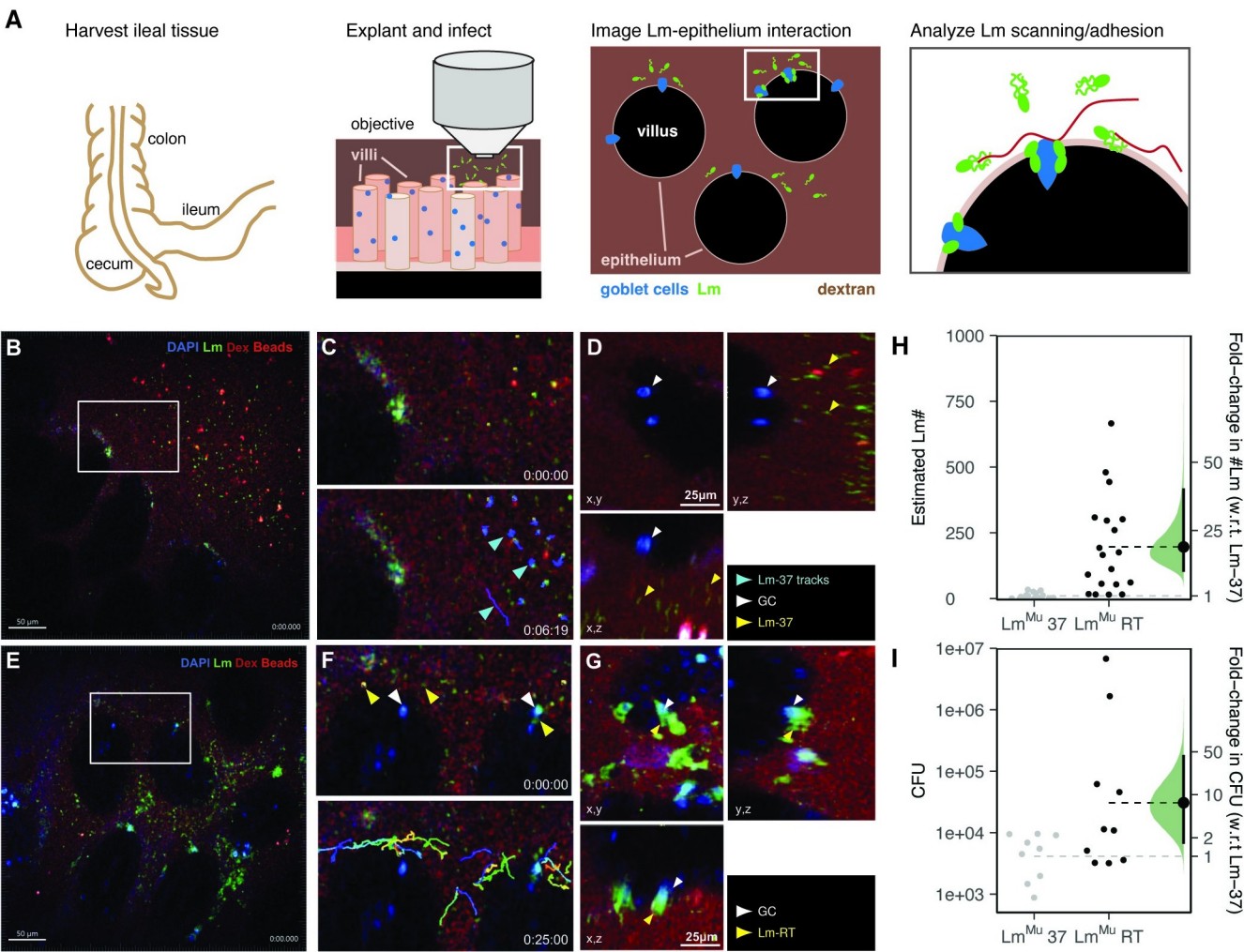

**Fig 2. Lm<sup>Mu</sup> invasion dynamics of mouse intestinal epithelium.** A: schematic of the experimental setup: mouse intestine (ileum) was explanted, challenged with Lm-RT or Lm-37, and imaged with time-lapse 2P microscopy to analyze scanning of the epithelium and interactions with goblet cells. B: images after challenge with Lm-37 (green), with C: zoomed views from a 25s recording showing that Lm-37 is predominantly non-motile (time encoded tracks, cyan arrows) and colocalize with fluorescent beads (pinkish red) trapped at the mucus interface. Rh-dextran (red) was added to confirm the integrity of the epithelium and label the fluid phase. D: 10min post infection (mpi), Lm-37 (green, yellow arrows) does not accumulate around secreting GCs identified by brightly stained nuclei (white-blue, white arrow). Right and lower panels are orthogonal views projected along the x and y dimensions respectively. E: Lm-RT (green, yellow arrows) is motile and penetrate the mucus layer to move along the surface of the epithelium. F: zoomed views, within minutes, Lm-RT (yellow arrows) can be seen scanning and adhering near GCs (white arrows). Bottom panel, Lm-RT time-encoded tracks (blue to red) show examples of Lm scanning the epithelium surface. G: 10 mpi, Lm-RT (green, yellow arrows) accumulates around GCs (bright white-blue nuclei, white arrows). Right and lower panels are orthogonal views projected along the x and y dimensions respectively. H: estimated # Lm invading for Lm-37 and Lm-RT was assessed by counting the number of green voxels that overlap with DAPI stained nuclei in the epithelium 10mpi. Invasion was about 19-fold higher (95% CI: 9.8–41) with Lm-RT. Each point represents an image (from 4 mice total). I: mice were infected orally with either Lm-37 or Lm-RT and colony forming units (CFU) were determined in the spleen 3 dpi (days post infection). 3 d.p.i., Lm-RT infection yielded 7-fold higher CFU compared to Lm-37 (95% CI: 1.6–45). Each point represents one mouse. In H,I, horizontal lines represent means for Lm-RT and Lm-37. On the right, plots show the fold-change from Lm-37 to Lm-RT (black dots), along with its bootstrapped distribution (green) and 95% CI (line segments).

web tool requires no prior knowledge or installation and allows readers to explore the model with different settings and export results.

The model simulates a 125x125μm patch of epithelium with periodic borders, such that cells crossing the border on the right will re-enter the field on the left and vice versa. Separate layers describe the dynamics of the phagocytes, the epithelium, and the Lm scanning it (Figs 4 and 5). Each layer contains its own CPM, essentially an image consisting of different "pixels",

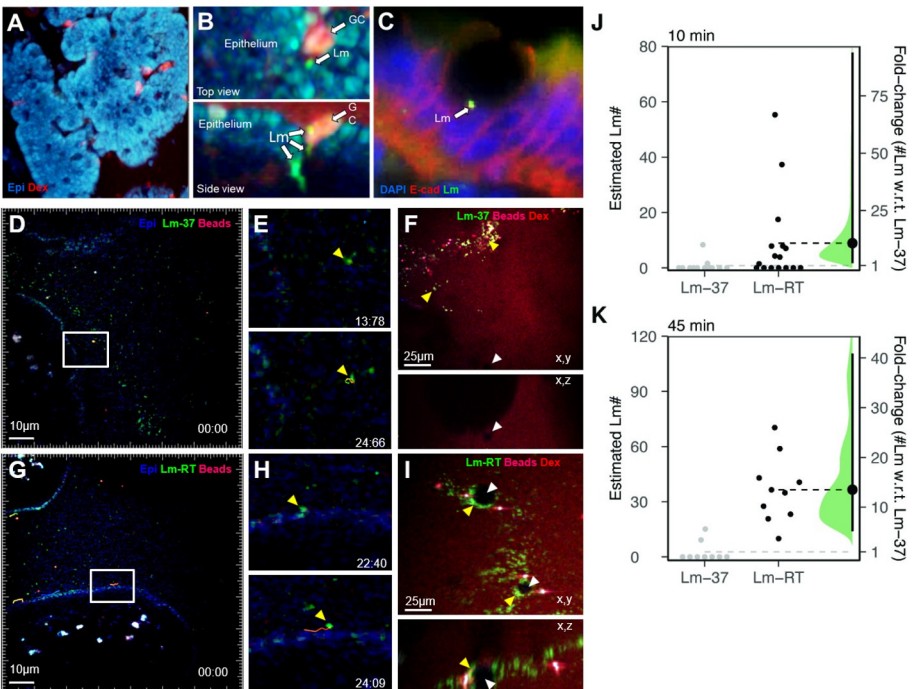

**Fig 3. Lm^wt interaction dynamics with human intestinal epithelium.** A,B: human intestinal resection specimens were challenged with Lm^Wt (green) and imaged with two-photon microscopy. A: autofluorescence of epithelial cells (blue) excited at 820nm. Several secreting goblet cells have been labeled by Rh-dextran (red) uptake. B: Lm-RT (green) can be seen binding to GCs and crossing the epithelium. C: Lm-RT (green) binding to E-cadherin (red) near a GC. Epithelial cell nuclei stained with DAPI (blue). D,E: human intestinal resection specimens were challenged with Lm and imaged with time-lapse two-photon microscopy. D: Lm-37 localizes predominately in the mucus layer identified by fluorescent beads (red) above the epithelium (blue, autofluorescence), with E: zoomed in views from a time-lapse sequence showing a Lm-37 track (time encoded). F: Lm-37 rarely attaches to the epithelium and fails to accumulate around GCs 45mpi. G: Lm-RT is often seen penetrating the mucus and scanning along the epithelium, with H: zoomed in views from a time-lapse sequence with an example track showing Lm-RT scanning the epithelium. Time stamp = min:sec. I: Lm-RT (green) attaches to the epithelium and accumulates around goblet cells (dark shadows) forming "rings". Lm invasion efficiency with Lm-37 and Lm-RT was assessed at J: 10mpi and K: 45mpi. Invasion was significantly higher for motile Lm-RT (fold-change w.r.t. Lm-37 was 11 at 10 mpi and 14 at 45 mpi, with 95% CIs [2.1–94] and [5.1–41], respectively). Bootstrapped fold-changes displayed as in Fig 2H and 2I; each point represents an image from at least 3 independent mice per condition.

reflecting bits of space that either contain a specific cell or only empty background. We chose a spatial resolution of 2 pixels/μm; at this scale, Lm typically occupy 1–2 pixels whereas epithelial cells and phagocytes are resolved in more detail (further increasing this resolution would only slow down the simulation unnecessarily). The model then changes over time in simulated "monte carlo steps" (MCS, here 1MCS = 1 sec).

Every MCS, cells first move within their own respective layers, modelled to match the motility of real cells (neutrophils/bacteria; based on published models [31, 32, 43] respectively; see "CPM dynamics", "Bacterial CPM", and "Phagocyte CPM" in S1 Methods for details). To account for the much higher motility of bacteria, the bacterial CPM was run for $v_{rel}$ =150 steps/s. After migrating *within* their respective layers, cells then interact *between* the layers, letting bacteria either: (1) be phagocytosed by overlapping phagocytes (rate $k_\varphi$), (2) attach to overlapping goblet cells (rate $k_{attach}$), or (3) fully "invade" the goblet cells they are attached to (rate $k_{infect}$). For further details on these processes, we refer to S1 Methods ("invasion and phagocytosis dynamics"). For an overview of model parameters and an explanation of how they were selected, please refer to S1 Methods).

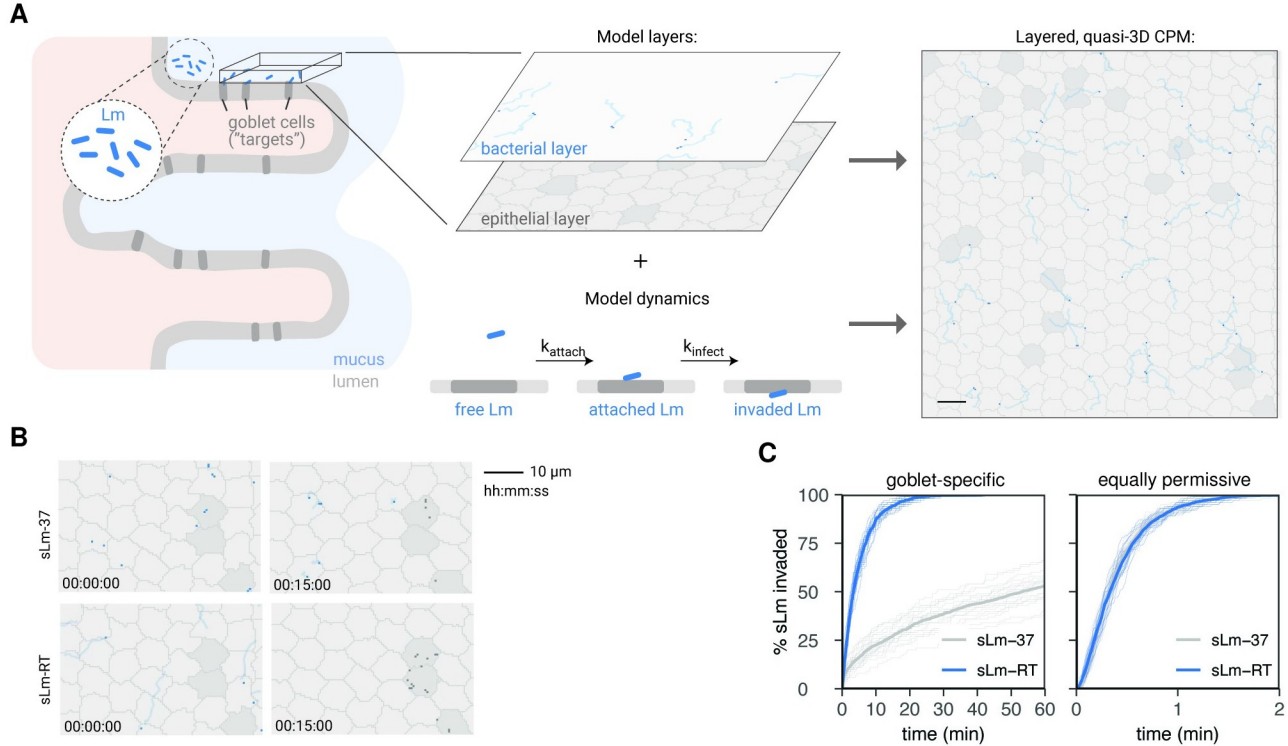

**Fig 4. Bacterial motility enhances target cell invasion in a cellular Potts model (CPM) of the epithelium.** A: The CPM describes the surface of the small intestinal epithelium (left). The model consists of two layers: the gut epithelium (including target cells in dark gray) and a layer above where bacteria (simulated Lm, sLm) move. Bacteria (blue dots) are shown with their traces (light blue). A bacterium that finds itself above a target cell in the epithelial layer can attach to it (rate $k_{attach}$), and subsequently invade (rate $k_{infect}$); both are irreversible processes. These layers together with the invasion dynamics yield the full, quasi-3D model of the gut epithelium (right; scale bar = 10μm). See Methods for details. B: Example screenshots of non-motile (sLm-37) and motile bacteria (sLm-RT) at the start of the simulation and after 15 min. C: % sLm invading over time, showing 20 individual simulations (thin lines) and average invasion (thick lines) for both non-motile (gray, sLm-37) and motile bacteria (blue, sLm-RT). Left: "goblet-specific" invasion where sLm can only invade goblet "target" cells, right: invasion when all epithelial cells are target cells (equally permissive to sLm).

## Simulation analysis

All simulations were performed for a total duration of 1 hour (=3600 MCS in the epithelial/ phagocyte models, and 3600*$v_{rel}$ steps in the bacterial model). Events of invasion and phagocytosis were tracked over time and further processed to obtain outcomes (%bacteria invaded/ phagocytosed and %target cells infected). These outcomes were plotted (using R) as mean ± standard deviation (SD) or standard error (SE) of 20 independent simulations. This number was sufficient that the SE was typically not or barely visible underneath the plotted lines, reducing uncertainty to a negligible level.

## Supplemental methods

Supplemental methods containing full details about the model construction and the selection of model parameters are available in S1 Methods.

## Results

### Murinized Lm cultured at RT is flagellated and highly motile

First, we evaluated the effect of culture temperature on flagella formation and motility of murinized Listeria. Lm was cultured in Brain Heart Infusion media (BHI) with shaking at room

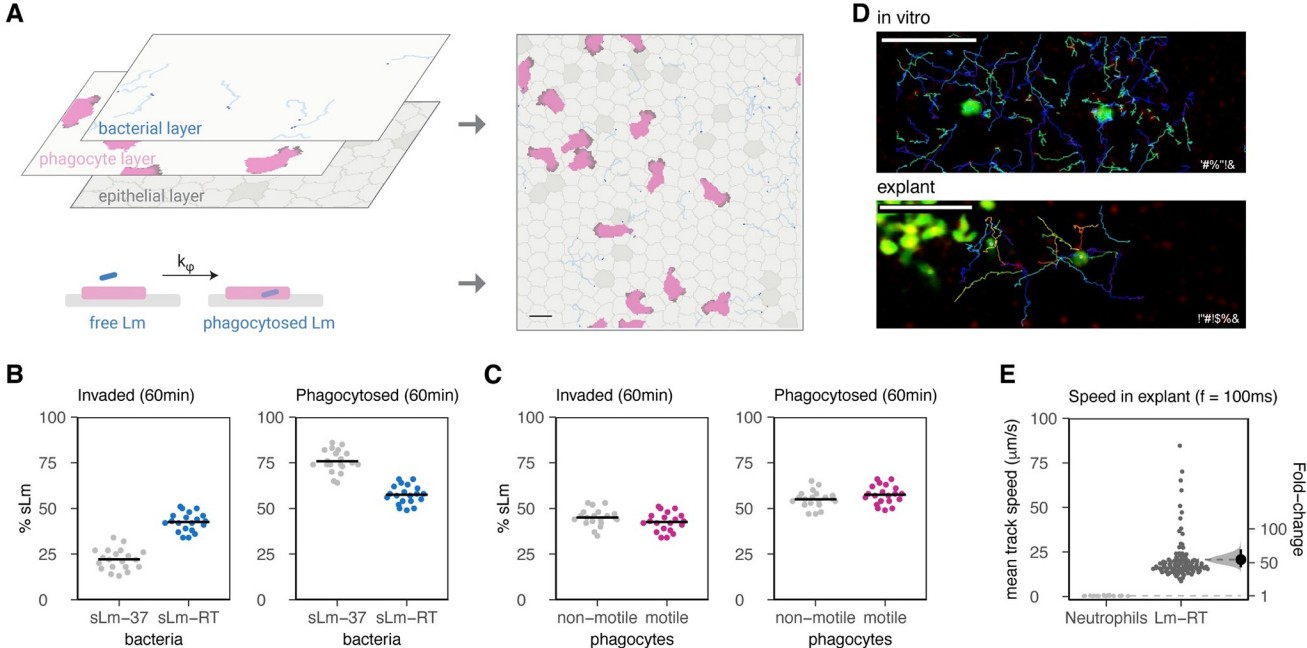

**Fig 5. Bacterial motility, but not phagocyte motility, drives Lm-phagocyte interactions at the epithelium.** A: To simulate bacterial phagocyte interactions on the epithelium, the model is extended by adding an extra layer containing migrating phagocytes (left). Phagocytes can phagocytose bacteria that move over them, or that have attached but not yet invaded a target cell. Phagocytosis is irreversible and occurs at rate $k_\varphi$. These layers and dynamics are again combined to obtain the full model (right; scale bar = 10μm). See also Fig 4 and S1 Methods for details. B: % sLm invaded (left) and phagocytosed (right) after 60 min for both non-motile (sLm-37) and motile (sLm-RT) bacteria. C: the same as panel B, but now comparing motile (pink) versus non-motile (gray) phagocytes, while bacteria are all motile (sLm-RT). Non-motile phagocytes were modelled by setting the parameters $\lambda_{act} = 0$, $max_{act} = 0$ (see S1 Methods). For B,C, horizontal bars indicate the mean of 20 independent simulations. D: To confirm neutrophil-Lm-RT interaction dynamics, both cell types were imaged both in vitro (imaging both neutrophils and Lm-RT, see Methods) and in vivo (in an explanted ileum after rechallenge with BacLight-Red-labelled Lm-RT, see Methods). Images show LysM-GFP neutrophils in green, together with time-encoded tracks of Lm-RT over several seconds (frame rate: 100 ms). On this time scale, neutrophils are essentially static while many Lm-RT can be observed moving at high speeds, often colliding with neutrophils and moving past them. Scale bars represent 50μm. E: comparison of mean track speeds of Lm-RT and neutrophils as determined in the in vivo system from panel D (frame rate: 100ms) Lm-RT move at least ∼55 times faster than neutrophils (95% CI [43–70]). Bootstrapped fold-changes (Lm-RT w.r.t. neutrophils) are displayed as in Fig 2H and 2I.

temperature (Lm-RT) or at 37˚C (Lm-37), and flagella formation was assessed by transmission electron microscopy. Lm-37 (Fig 1A and 1B) was non-flagellated while Lm-RT showed multiple long peritrichous flagella, similar to published findings with the wild type strain EGD Lm$^{Wt}$ [22], confirming that RT culture produces flagellated murinized Lm bacteria.

We next assessed bacterial motility in vitro using video microscopy. Because EGD Lm and Lm$^{Mu}$ were nearly the same in terms of motility (S1A and S1B Fig), we focused our analysis on EGD Lm as it is more clinically relevant. Lm-37 and Lm-RT were plated on slides for time-lapse imaging and cell motility was tracked (red dots, Fig 1C and 1D, S1 and S2 Movies). In contrast to Lm-37, which was predominantly non-motile (<2% motile, Fig 1C), Lm-RT was highly motile with heterogeneous speeds and long persistent tracks, moving in random directions across the surface of the slide (Fig 1D and 1E, S1 and S2 Movies). Basic motility parameters for Lm-37 and Lm-RT were calculated and compared (Fig 1E–1I) using *celltrackR* [38]. Despite large variation in average track speed, Lm-RT motility (∼10 μm/s) dramatically exceeded that of Lm-37, (which was essentially non-motile; <1μm/s Fig 1E and 1F). These data confirmed that Lm-RT is highly motile with strongly increased mean squared displacement (motility coefficient M = 19 μm²/sec, S1C–S1F Fig compared to Lm-37, M = 0 μm²/sec).

Lm-RT cultures also contained a subset of non-motile or "spinning" bacteria that were stuck to the glass slide. These tracks did not represent motility behavior per se but rather were an artefact of our in vitro system. Therefore, we filtered out these tracks to obtain an idealized "RT-motile" (RT-m) population for further analysis (S2 Fig). In contrast to the non-motile Lm-37, Lm-RT(-m) tracks were well-described by a persistent random walk model (S1 Fig) with a motility coefficient M = 19 μm$^2$/s in the total Lm-RT population. This motility coefficient increased more than 2.5-fold (M = 52 μm$^2$/sec) upon filtering for the motile (RT-m) subset (S1A and S1C Fig), contributing to even larger displacements over time (Fig 1G). RT-m also had a (slightly) higher persistence time (P = 0.7s vs P = 0.43s or 1.12s vs 0.99s, depending on the analysis method; see S1C–S1F Fig). In line with such short-term persistent motion, average turning angles were below 90˚ for time intervals up to two or three seconds, with lower turning angles observed in the RT-m subset (Fig 1H). This directional autocorrelation was not due to global directed motion or local alignment due to crowding: average angles between cells were (nearly) equal to 90˚ (Fig 1I), suggesting that individual bacteria moved in directions independent of that of other bacteria. In summary, these data show that Listeria cultured at RT display a high, persistent random walk-like motility. Importantly, Lm-RT maintains this motility even after several hours of subsequent culture at 37˚C (S1G Fig, with 79%, and 56% motile bacteria after 2h and 3h culture at 37˚C, respectively). These findings suggest that ingested Lm would remain motile in vivo as they reach the small intestine and invade the epithelium.

## Motile Lm arrives at the intestinal epithelium after oral infection in vivo

To confirm that Lm-RT retains motility in vivo long enough to reach the small intestine (specifically, the ileum), we orally infected mice with human Lm-RT (EDG-GFP), which binds poorly to mouse E-cadherin and thus allowed us to assess motility separate from E-cadherin mediated attachment. Indeed, 2P microscopy revealed many examples of robust Lm-RT motility in the ileum 1–1.5 hours after oral infection (S3 Movie and S3 Fig). The presence of Lm in the ileum at 1hpi is consistent with gut transit times in mice as measured using technetium-labeled activated charcoal and fluorescent dextran [44, 45]. Together, these data show that after oral infection, Lm-RT reaches the ileum and interacts with the surface of the epithelium in a motile state.

## Motility facilitates Lm scanning of the intestinal epithelium and enhances invasion

Next, we hypothesized that increased motility may help Lm-RT locate potential sites of invasion (goblet cells). Even if motility is random (with no specific directionality towards goblet cells), motile Lm would still be expected to encounter goblet cells more frequently. We therefore examined the bacterial behavior at the epithelial surface using explanted intestinal tissues and 2P microscopy. Sections of B6 mouse intestine (ileum) were harvested, glued to plastic coverslips, and then sliced open to expose the epithelial surface. Intestines were challenged with murinized Lm-37 or Lm-RT and time-lapse recordings collected by 2P microscopy to assess bacterial motility, epithelial adhesion, and invasion (Fig 2A, S4 and S5 Movies). Non-motile Lm-37 was predominantly found collecting on the top of the mucus layer (10–40 μm above the epithelium) along with red fluorospheres that served as a control to identify regions of the intestine were mucus was thin or absent (Fig 2B–2D). In contrast, motile Lm-RT often penetrated the mucus layer within minutes and multiple bacteria were observed scanning along the epithelium (Fig 2E–2G).

Even in regions where mucus was thin or absent, Lm-37 rarely contacted or attached to the epithelium. However, in intestines challenged with Lm-RT, bacteria accumulated on the epithelium in clumps associated with DAPI stained nuclei, presumably goblet cells—which were shown previously to stain brightly with DAPI during secretion and antigen transcytosis [37] and are the preferred target cell for invasion [12]. This suggests that Lm-RT motility—be it directed or random—facilitates interactions with goblet cells compared to non-motile Lm-37.

To further quantify Lm attachment to the epithelium, we measured BacLight-Green stained bacteria that bound to the epithelium (green voxels) over time (Fig 2H). Few bacteria were found attached to the epithelium initially after challenge. However, within 15min, Lm-RT showed a 19-fold higher attachment compared to Lm-37, which attached infrequently (Fig 2H, fold-change 95% CI: 9.8–41). We also performed CFU assays on orally infected mice to assess whether motile Lm-RT infects mice more readily in vivo than non-motile Lm-37. We found that mice infected with Lm-RT had 7-fold higher bacterial burdens in the spleen 3 days after infection than mice challenged with Lm-37 (95% CI: 1.6–45, Fig 2I). Thus, even though higher temperatures are associated with an upregulation of virulence factors that would be expected to *increase* infectivity [46–48], we instead found that motility provided an infection advantage (similar to O'Neil et al [22]). These results show that Lm-RT actively scans the mouse intestinal epithelium to efficiently locate and attach to goblet cells, and that this motility is associated with increased infection.

## Motility promotes Lm interactions with human epithelium and invasion

Mouse infection models provide important in vivo insight into bacterial pathogenesis but, due to the species specificity, standard oral models using Lm$^{WT}$ are inappropriate for studying early host-pathogen interactions in the gut. InlA murinized Lm strains have been used to study oral infection, but murinized Lm can bind mouse N-cadherin in addition to E-cadherin, potentially affecting invasion specificity [17]. Therefore, in a complimentary approach, we examined Lm$^{WT}$ invasion in explanted human intestinal tissue biopsies. Human tissue explant systems have the advantage of preserving species-specific invasion mechanisms and host-pathogen interactions [49].

We used 2P microscopy to assess whether Lm$^{WT}$ motility impacted epithelial invasion in explanted human ileum. Fresh surgical biopsy specimens were collected from the Washington University DDRCC (Digestive Disease Research Core Center). The muscle layer was removed, and tissue samples were placed in a custom imaging chamber to hydrate the tissue with warm oxygenated media. Tissues were infected with either EGD-GFP Lm$^{WT}$ grown at 37°C or RT for time-lapse imaging (Fig 3, S6 and S7 Movies). The epithelial barrier of explanted human intestine maintained its integrity for several hours as shown by the exclusion of Rh-dextran from beneath the epithelium (Fig 3A). Human epithelial cells are brightly auto-fluorescent when excited with 800–850nm 2P laser light (Fig 3A), and goblet cells often appear red due to the uptake of soluble Rh-dextran following mucus secretion, a phenomenon called goblet-cell associated antigen passages [37]. EGD-GFP bacteria co-localized with goblet cells and were found immediately below the epithelium where lamina propria phagocytes reside (Fig 3B). Human ileum infected with EGD and stained with antibodies to E-cadherin shows EGD-GFP bound to adherens junctions deep in the epithelium near cells with goblet cell morphology (Fig 3C).

We also infected intestinal explants with EGD-GFP Lm and analyzed their behavior and interactions with the intestinal mucosa (Fig 3D–3I). Like in the mouse epithelium (Fig 2), non-motile EGD-GFP Lm-37 primarily became trapped in the mucous layer identified by fluorescent beads (Fig 3F). However, in some areas where the mucus layer was thin or absent,

EGD-GFP Lm-37 drifted down to the epithelium and attached. In contrast, when intestine samples were challenged with EGD-GFP Lm-RT, we observed motile bacteria penetrating the mucus and approaching the epithelium or moving into regions where the mucus was discontinuous (Fig 3G–3I). When motile bacteria encountered the epithelium, they either bounced off and left the field of view or scanned along the epithelial surface (Fig 3H). We observed several examples of scanning bacteria suddenly arresting and adhering to the epithelium at locations where other bacteria had previously bound. Over 20–30 minutes, bacteria accumulated into green clumps in the epithelium (Fig 3I), presumably as Lm bound to E-cadherin on goblet cells. Indeed, staining with the goblet cell marker Cytokeratin-18 confirmed that bacteria accumulated near goblet cells (S4A Fig).

We quantified the extent of epithelial attachment by measuring the number of green pixels associated with the epithelium at 10 and 45 minutes after challenge and extrapolating the data to estimate bacterial numbers (Fig 3J and 3K, see Methods). EGD-GFP Lm-RT accumulated significantly more on the epithelium compared to EGD-GFP Lm-37 (Fig 3K). To confirm that this difference was due to Lm-RT motility, and not temperature-dependent changes in virulence factor expression, we performed 2P imaging experiments with a 10403 flaA deletion mutant [23] (Gift of Lisa Gorski, USDA) grown at RT. Indeed, flagella-deficient Lm grown at RT had profound defects in motility, mucus penetration and epithelial adhesion, similar to those observed in murinized 10403s and wild-type EGD when cultured at 37°C (S4 Fig and S8 and S9 Movies).

These results confirm that Lm-RT has an advantage over Lm-37 in human intestine, at least in the crucial first step of locating and attaching to goblet cells. This advantage is a direct result of the temperature-dependent change in motility.

## A layered cellular Potts model shows that bacterial motility can facilitate invasion by driving rapid interactions with target cells

Results in both the human and murine intestinal explant systems suggest that Lm motility enhances epithelial invasion of "target" cells (i.e., goblet cells). However, in the explant system it is difficult to separate the effect of increased mucus penetration from that of increased scanning along the epithelium, nor is it possible to investigate the consequences of motility over larger spatiotemporal ranges. We therefore used a computational biology approach to estimate how bacterial motility affects relevant outcomes over time, such as the number of bacteria that invade target cells or are phagocytosed over time. This allowed us to investigate which host-pathogen interaction parameters determine epithelial invasion efficiency.

We turned to the cellular Potts model (CPM), which can model complex tissues in space and time with realistic cell morphology, motility, and cell-cell interactions [30–32, 50–52]. Our "layered" L-CPM models bacterial motility and invasion dynamics at the epithelium, using simulations with aligned coordinates to simulate the nearly-2D system of bacteria scanning on top of the epithelial surface (Fig 4A and S10 Movie). This approach preserves the relative simplicity of 2D models while also allowing cells to move on top of each other and interact in a "quasi-3D" setting. Importantly, additional CPM layers can easily be added to model more complex interactions, for example adding host phagocytes to simulated host immune response dynamics at the epithelium (see next section).

Our L-CPM is constructed using data obtained from in vitro and in vivo experiments, which give it physiologically relevant spatiotemporal scales as well as quantitative parameters for bacterial motility, invasion efficiency and phagocyte dynamics. First, we measured epithelial cell diameters in vivo using 2P imaging of E-cadherin-CFP (cyan fluorescent protein) mice (gift of Charles Parkos and Ronen Sumagin) and estimated the prevalence of goblet cells (dark

goblet shaped cells) in the villus epithelium. These data were used to create the 2D topology of the epithelial layer in the L-CPM (Fig 4A). In addition, we derived a stereotypical bacterial motility behavior from the in vitro Lm-RT-m motility data (Fig 1) and chose model parameters to mimic this behavior in a "simulated Lm-RT" (sLm-RT) population (see S1 Methods and S5 Fig), while also simulating a non-motile sLm-37 population. Finally, we viewed time-lapse recordings of motile Lm near goblet cells in the explant intestine models (S4–S7 Movies) to estimate a physiological range for the rates of attachment and invasion in the CPM (Fig 4A, see S1 Methods for details).

We then used the CPM to simulate invasion dynamics of sLm-RT as described above (S10 Movie). Although Lm-37 was non-motile in vitro, we expect that in vivo, non-motile bacteria would diffuse along the epithelium rather than remain perfectly stationary. Therefore, we modelled sLm-37 with a slight diffusive motion to mimic this behavior and ensure that comparisons with sLm-RT are as conservative as possible (S10 Movie). Simulations consistently showed that when sLm has to scan for target cells, sLm-RT motility vastly sped up invasion dynamics compared to the diffusive motion of sLm-37 (Fig 4B and 4C). Most sLm-RT invaded the epithelium within 20–30 min (consistent with observations in Figs 2 and 3 and in line with published findings [12, 19]), while in the sLm-37 case, only 50% of bacteria had invaded as late as the 1-hour mark ($\sim$15x later than sLm-RT). The increased invasion efficiency of sLm-RT was a direct consequence of bacterial motility driving interactions with "target" cells (i.e. goblet cells), disappearing in simulations where all epithelial cells were permissive to invasion (Fig 4C). Indeed, increasing sLm speed facilitated invasion, (S6 Fig), but once sLm moved fast enough to reach target cells within the 60-minute timeframe, further increasing bacterial speed to "super physiological" levels did not confer any additional invasion advantage.

The finding that bacterial motility confers an invasion benefit is not trivial. While motile bacteria would be expected to find target cells more rapidly, they also, on average, spend less time in contact with the target cells. Therefore, motility could in in theory be *detrimental* to invasion in situations where the rate of target cell attachment is low. However, this detrimental effect did not occur even in simulations with very low attachment rates far outside of the physiological range (S7 Fig; the 1-hour invasion was consistently about 2-fold higher in sLm-RT than in sLm-37 across all attachment rates). Thus, our model predicts that for physiological bacterial speeds and attachment kinetics, motility helps bacteria locate and invade target cells more efficiently.

## In ongoing infections, neutrophils meet Lm on the apical epithelial surface

Next, we examined how bacterial scanning of the epithelium might be affected by host cellular immunity. In response to intestinal infection, neutrophils extravasate from submucosal vessels and migrate through the lamina propria towards sites of infection [40]. Neutrophils can patrol the epithelium directly from the basolateral side as well as undergo transepithelial migration to patrol the apical surface [53, 54]. However, the exact role of neutrophils on the apical side of the epithelium remains unclear.

We hypothesized that after the first few hours of infection, neutrophils on the apical surface of the epithelium could potentially reduce invasion and initial bacterial burden by phagocytosing bacteria that are scanning or attached to the epithelium. While it is true that neutrophil recruitment takes several hours and cannot prevent invasion events entirely, natural infections typically involve an ongoing supply of bacteria rather than the single challenge used in most mouse models (for example, when a food source or water supply is contaminated). In such a scenario, pathogenic bacteria may continue to pass through the digestive tract for hours or days and the epithelium after neutrophils have already undergone transepithelial migration to

the luminal surface. Yet it remains unclear whether neutrophils indeed interact with bacteria and can limit further infection through phagocytosis.

First, we used 2P imaging to examine whether neutrophils were recruited onto the epithelium and observed substantial numbers of neutrophils patrolling the luminal surface of the epithelium 5hpi (S8 Fig). Next, we simulated an ongoing infection by adding a second challenge of Lm^Mu-RT (BacLight-Red labeled), and analyzed Lm-RT motility and interactions with neutrophils (S8D Fig). These results confirm that transmigrating neutrophils on the epithelium can indeed interact with bacteria *before* they invade—and in some cases phagocytose them (S8E Fig).

## Bacterial motility, not phagocyte motility, drives outcomes in the L-CPM

The finding that neutrophils are recruited to the apical side of the epithelium following Lm infection suggests that they might have a protective function in clearing bacteria at the luminal surface during ongoing infections. However, it remains unclear how important these interactions are for preventing further bacterial invasion—or how Lm motility affects these host-pathogen interactions at the epithelium.

To examine the impact of phagocytosis on bacterial invasion efficiency, we added phagocytes to the L-CPM and simulated phagocyte-bacteria interaction dynamics. We tracked and quantified the in vivo motility of LysM-GFP neutrophils in 2D as they crawled along the epithelium in 2P microscopy recordings (S11 Movie and S9 Fig). To simulate host pathogen interactions at the epithelium, we expanded our L-CPM by adding a layer of phagocytes with realistic cellular morphology, dynamics and migration behavior based on our in vivo data (Fig 5A and S10 Fig, see Methods). In this L-CPM, phagocytes can "hunt" sLm as they are scanning the epithelium or immediately after they have attached to target cells, but not yet invaded (see Fig 4A).

Although Lm infection induces robust neutrophil recruitment, reliable measures for the phagocytosis rate are currently lacking. Furthermore, the number of neutrophils present on the epithelium is highly context-dependent and may vary over time and in different regions. Since our focus is understanding how bacterial motility affects host pathogen interactions, we started from a simple, "equal-opportunity" baseline where phagocytes covered roughly the same surface area as target cells and the phagocytosis rate equaled the rate of sLm attachment to target cells (Fig 5, ensuring that the relative rate differences themselves did not predetermine invasion and phagocytosis outcomes). However, since the true values of these parameters are unknown, we later vary them systematically to assess how they affect our conclusions (see below).

While invasion was again reduced about 2-fold with sLm-37 compared to motile sLm-RT in these baseline simulations, the opposite was true for phagocytosis which was ∼30% more efficient with non-motile sLm-37 than sLm-RT (Fig 5B). Non-motile sLm-37 are inefficient at finding (also non-motile) target cells to invade. They would be equally inefficient at finding phagocytes but since phagocytes are themselves motile, they can still find and sweep up non-motile bacteria from the epithelium to phagocytose (S12 Movie). By contrast, phagocytes lose this advantage when faced with the rapidly motile sLm-RT; motile bacteria rapidly encounter both static target cells and slowly moving phagocytes but are also more likely to "escape" from these encounters before invasion or phagocytosis is completed (S12 Movie). Thus, motility always helps sLm find other cells faster (be it target cells or phagocytes) but the success of these encounters depends on whether they last long enough to facilitate the attachment or phagocytosis process.

Because our simple baseline scenario uses "best guesses" for unknowns such as phagocyte numbers and phagocytosis rate, we next examined more thoroughly how sLm fate depends on

these parameters. We first varied the prevalence of sLm, target cells, and phagocytes (S11 Fig and S13–S15 Movies). Again, invasion at 1 hour was consistently about 2-fold higher for motile sLm-RT than for sLm-37. Although the absolute percentages of invasion and phagocytosis varied, the 2-fold difference between sLm-RT and sLm-37 was observed for all phagocyte prevalences (S11I Fig) and most challenge doses (S11A Fig), suggesting it would hold no matter how many neutrophils transmigrate to the apical surface of the epithelium. The magnitude of this motility-dependent invasion benefit only varied with the number of target cells (S11E Fig)—consistent with the hypothesis that motility helps Lm find target cells when these are rare.

While challenging the epithelium with more sLm mostly did not alter the fraction of sLm invaded or phagocytosed, additional target cells or phagocytes shifted Lm fate towards more invasion or phagocytosis, respectively (S11 Fig and S13–S15 Movies). Indeed, when sLm can either invade or be phagocytosed, target cells and phagocytes essentially compete for bacteria; for motile bacteria, it is then the relative surface area of target cells and phagocytes that determines which type of interaction partner it most likely encounters first (S11D, S11H and S11L Fig). Thus, while motile bacteria always invade faster than non-motile counterparts, this invasion can still be mitigated if sufficient phagocytes are present to intercept them. Even a few phagocytes can reduce sLm invasion substantially (S11L Fig).

Increasing the rates of target attachment or phagocytosis similarly shifted the invasion-phagocytosis balance and in contrast to cell prevalences, the choice of these rates *did* affect the strength of the motility-dependent invasion benefit of sLm-RT (S12 Fig). Still, sLm-RT retained its invasion benefit over sLm-37 in most cases, as this trend was reversed only in cases where phagocytosis was >3x faster than attachment. Thus, sLm motility drives both target cell and phagocyte encounters and almost always confers an invasion benefit, but both this benefit and the invasion-phagocytosis balance depend on quantitative parameters (such as numbers of target cells and phagocytes present, and the relative rates of attachment and phagocytosis). Nevertheless, we conclude that motile bacteria retain their invasion benefit over non-motile bacteria in the presence of phagocytes unless phagocytosis is over three times as fast as bacterial attachment and invasion.

Finally, a general but striking conclusion followed from this model: bacteria move so much faster than phagocytes that phagocytes are fundamentally incapable of "hunting" motile bacteria whether that is through random migration or chemotaxis. Rather, phagocytes rely on bacterial motility to drive capture and phagocytosis (S12 Movie). Indeed, while sLm motility strongly affected invasion and phagocytosis (Fig 5B and S13A–S13C Fig), *phagocyte* motility did not, except when sLm was also non-motile (sLm-37, Fig 5C and S13D–S13F Fig). This prediction was tested in vitro by imaging blood neutrophils interacting with motile Lm-RT. 2P microscopy corroborated our prediction by showing that indeed, neutrophils migrate much too slow to "hunt" motile Lm directly (Fig 5D and S16 Movie). Furthermore, we imaged neutrophil and Lm-RT interactions in explanted mouse ileum and once again found a massive speed difference between Lm-RT and neutrophils, with Lm moving ~50x faster than nearby neutrophils on the epithelial surface (Fig 5D and 5E and S17 Movie). Thus, both our model and data suggest that once phagocytes have reached the epithelium, their motility will have negligible impact on infection outcomes with bacterial pathogens that are as motile as Lm.

## Discussion

Many important human bacterial pathogens are highly motile (Lm, *E. coli*, *P. aeruginosa*) [25, 26, 55], yet the impact of bacterial motility on infection remains incompletely understood [56]. We addressed this question for Lm and found that Lm motility may enhance epithelial

infection by allowing bacteria to efficiently scan the epithelium for target cells that are permissive to infection. This hypothesis is based on the observation that Lm preferentially invades the host via goblet cell transcytosis [12] and that human food-borne infections typically result from Lm living at low temperatures (≤RT), which is flagellated and motile. Indeed, we confirmed that Lm-RT is flagellated and highly motile and showed that its motility resembles a random walk with strong directional persistence.

## Lm bacteria are motile when they reach the gut epithelium

Most human infections result from eating contaminated foods (stored at less than 37˚C), yielding bacteria that are presumably flagellated and motile when ingested. An important question is therefore whether bacteria preserve this motility long enough to traverse the gastrointestinal tract and scan the intestinal epithelium.

One option is that Lm could rapidly downregulate flagella expression after ingestion and would therefore enter the small intestine in a non-flagellated state. Indeed, flagellin is a potent TLR ligand, and lack of expression by Lm-37 could confer an infection advantage by making Lm less immunogenic. However, we found that Lm-RT (both EDG and 10403s strains) retains motility for several hours after incubation at 37˚C (79% motile at 2h and 56% at 3h, S1E Fig). Furthermore, we were able to directly confirm that motile Lm is present in the murine ileum as soon as 1 hour after oral gavage (S3 Fig). We used EGD-GFP Lm for these experiments since it does not bind mouse E-cadherin, and allows us to assess motility in the absence of E-cadherin mediated attachment and invasion. Using this approach, we could confirm directly that Lm remains motile as it traverses the small intestine—raising the question how this motility affects interactions with the intestinal epithelium and pathogenesis.

## The role of motility in bacterial pathogenesis

The role of flagellar motility has been extensively investigated in the context of Salmonella pathogenesis. In an oral infection model (newly hatched chicks), non-flagellated mutant Salmonella was found to be less virulent in vivo, but using a gut explant approach, the authors concluded that this was likely due to flagella mediating epithelial adhesion rather than motility per se [57, 58]. In contrast, when the same Salmonella mutant strain was examined in a rat ileal explant model, motility enhanced infection, presumably by increasing bacterial interactions with the intestinal epithelium [27]. Likewise, Barbosa et al. found that chicks infected by gavage with non-motile (ΔmotB) and non-flagellated (ΔfliC) Salmonella generated significantly lower CFU 3–5 dpi in the cecum compared to wild-type Salmonella Enteritidis [28]. The importance of motility to Salmonella infection was further elucidated in a fascinating study by Furter et al., who demonstrated with live microscopy of explanted mouse colon and cecum that motility allowed Salmonella to probe the mucus barrier for defects and access the epithelium [29].

While a beneficial effect of motility on invasion has been observed for Salmonella, similar studies on the role of Lm motility in infection have been inconsistent. Way et al. found that bacterial burdens and immune responses were similar between Lm flagellar null mutants and WT bacteria in both i.v. and oral gavage infection mouse models [59]. In other work, however, ONeil et al. reported that infection with flagellated Lm outcompeted non-flagellated bacteria 6–16 hours after oral challenge [22]. Experimental systems could account for these different findings, since mice infected by i.v. injection bypass the epithelial invasion step, and infection by gavage can produce small tears in the esophagus and lead to direct entry of Lm into the circulation in contrast to oral ingestion approaches [15, 60].

In our current study, we used a combination of different model systems to show that motile Lm more readily penetrates the mucus barrier above villi and gains access to the epithelial surface of the ileum. This finding resembles the results in the Salmonella model [29], although the explanted ileum has a less dense mucus layer compared to the colon studied there. Indeed, future iterations of the L-CPM model could incorporate a simulated mucus layer to investigate how motile bacteria interact with the mucus, which is relevant to understanding bacterial pathogenesis as well as the colonization of healthy microbiota.

However, beyond facilitating mucus layer penetration and access to the epithelium, motility has an additional effect in the case of Lm: given the distinct cellular tropism for invasion (focused at goblet cells), we found that motility provided a crucial advantage by promoting epithelial scanning to locate invasion "targets".

## The effect of temperature on Lm motility and pathogenesis

One important caveat to our study with Lm-RT and Lm-37 is that culture temperature affects more than just flagellin expression and bacterial motility. Lm virulence genes are silenced at low temperatures but dramatically upregulated by PfrA [46, 47] expression at 37˚C. In groundbreaking work from the Cossart group, temperature was shown to regulate PrfA expression via changes in the mRNA secondary structure, which blocks ribosome entry at lower temperatures but allows PrfA expression at 37˚C to upregulate Lm virulence factors [48]. This mechanism is separate from the downregulation of flaA expression at 37˚C, which is mediated by the repressor MogR [20]. A MogR deletion mutant was used to demonstrate that flaA downregulation is crucial for cell-cell spread and full virulence in vivo [20].

Importantly, the EGD Lm strain used in this study has a mutation in PrfA, allowing it to express virulence factors constitutively independent of temperature—raising the question to what extent we can compare the RT and 37˚C conditions in this strain. Nevertheless, we believe this comparison is valid for several reasons.

First, we found that the PrfA mutation did not negatively regulate EGD motility, which was equivalent to that of the murinized Lm 10403s strain without a PrfA mutation (in line with the idea that motility is regulated through a separate mechanism). Second, while the constitutive expression of virulence factors in EGD could still enhance the virulence of EGD Lm-RT, we would expect a similar effect on Lm-37, suggesting that any differences between EGD RT and 37˚C are independent of the PrfA mutation. Third, the virulence of different Lm strains has been evaluated carefully by Becavin et al., and the EGD PrfA mutation was found to affect host cell invasion at low temperatures (30˚C) only but not in vivo [33]. Importantly, our experiments were performed under conditions (i.e., 37˚C for explant imaging and in vivo CFU assays) where the PrfA mutation is not expected to affect the results or conclusions of our study. Fourth, our experiments with murinized Lm (10403s strain) clearly showed Lm-RT had an invasion advantage over Lm-37, mirroring the results with EGD and human ileum.

Finally, to directly address the possibility that temperature dependent virulence factor expression might confound our experimental results, we performed 2P imaging experiments with a 10403 flaA deletion mutant [23] (Gift of Lisa Gorski, USDA) grown at RT and showed that it had profound defects in motility, mucus penetration and epithelial adhesion, similar to 37˚C cultured murinized 10403s and wild-type EGD (S4 Fig). Thus, we are confident that flagellar driven motility, rather than virulence factor expression, is the key factor driving mucus penetration, epithelial scanning, and invasion in our temperature shifted Lm experiments.

## Bacterial scanning of the epithelium facilitates invasion

2P imaging of explant tissues directly revealed that when motile Lm reached the gut epithelium, their motility allowed bacteria to penetrate the mucous layer and scan the surface of the epithelium. The question remains how this scanning might affect Lm pathogenesis.

In complementary computer simulations, we therefore integrated quantitative measurements of cell sizes and motility obtained from imaging experiments and assessed invasion outcomes over a wide range of infection scenarios (e.g. bacterial challenge doses and motility parameters) and time scales that are not feasible to image directly. Moreover, these simulations allowed us to disengage the effects of motility on epithelial scanning from the effect on mucus penetration.

Consistent with our two-photon imaging results, CPM simulations predict that bacterial motility enhances invasion by allowing bacteria to scan the epithelial surface and quickly locate preferred target cells for invasion. This invasion advantage associated with bacterial motility holds in most, but not all conditions tested—for example, it disappears when cell specificity is irrelevant (i.e., all epithelial cells are equally permissive for invasion), or when phagocytosis is so efficient that motile bacteria are phagocytosed before they can invade (very high rates of phagocytosis). Nevertheless, our model predicts that the beneficial effect of scanning is surprisingly robust: in absence of phagocytes, invasion at 1 hour was consistently $\sim$2-fold higher for motile bacteria, regardless of the rate by which they attach to target cells. Upon arrival of phagocytes, the size and direction of the effect does depend on attachment and phagocytosis rates—but it did persist at least qualitatively in almost all scenarios. Thus, unless there is reason to believe that phagocytosis rates are at least 3-fold higher than the rate of Lm attachment to target cells, our model predicts that motile bacteria have an invasion advantage once they reach the epithelial surface.

## Bacterial motility, not phagocyte motility, drives phagocytosis

Aside from showing how bacterial motility facilitates invasion, we used our models to examine how patrolling neutrophils on the luminal surface of the epithelium might affect bacterial invasion. While this is unlikely to occur in the case of a single bacterial challenge (where neutrophils would be recruited hours after bacteria reach the gut), we reasoned that in an ongoing infection, newly arriving bacteria might encounter phagocytes while scanning the epithelial surface. In luminal rechallenge experiments, we did in fact observe motile Lm interacting with neutrophils (S17 Movie)—but these interactions were driven by bacterial motility rather than neutrophil patrolling.

Indeed, our quantitative results challenge the fundamental paradigm of phagocytosis as "predators". By simulating bacterial and phagocyte dynamics to scale, our CPM clearly shows that neutrophils cannot possibly "hunt" motile bacteria—which move 1–2 orders of magnitude faster. We estimated this speed difference to be $\sim$120X based on in vitro Lm tracking data (Fig 1) and in vivo neutrophil motility (S9 Fig). A direct estimate based on ileal explant imaging was slightly lower ($\sim$50x, Fig 5E), but this is likely an underestimate because tracking errors inflate neutrophil speeds estimated from 100msec recordings. Regardless, even a conservative 50x speed difference between phagocytes and bacteria would be equivalent to a human ($\sim$10mph or 16 km/h) running after a commercial passenger jet ($\sim$560mph or 900 km/h).

In contrast to phagocyte motility, bacterial motility *can* drive an increase in phagocytosis in our CPM—at least in conditions where phagocytosis is more efficient than invasion. This suggests that rather than pursuing their prey, phagocytes may act more like "fly paper", capturing and ingesting motile bacteria in the local environment through random collisions. Indeed, goblet cells and phagocytes can be seen as two competing traps, whose relative "surface area"

(abundance of goblet cells/phagocytes) and "stickiness" (rates of goblet attachment/phagocytosis) determine the balance between attachment/invasion and phagocytosis.

We note that even though phagocyte motility had a negligible role in capturing motile bacteria in our simulations, this does not suggest that motility is dispensable in host defense. Clearly, phagocyte motility is required for many aspects of phagocyte function including extravasation from blood vessels, trafficking to sites of infection, and the capture of non-motile bacteria. Our simulations showed that non-motile bacteria *can* be "hunted" actively by phagocytes, a scenario relevant to skin infections with S. aureus or S. pyogenes [61, 62] (both non-motile cocci), where phagocytes are important for controlling bacteria growth and dissemination through phagocytosis and cytokine secretion. Thus, our results do not contradict existing literature on the importance of neutrophil chemotaxis towards bacteria [61, 63, 64] – rather, they highlight that in the specific case of highly motile bacteria, neutrophil chemotaxis would not likely enhance capture and phagocytosis of individual bacteria once neutrophils have found their way to the site of infection.

## Future directions

Our L-CPM allows us to investigate a wide range of infection scenarios by directly controlling the number of phagocytes and bacteria in the simulations over time. However, like any model, it has its limitations. First, it is worth noting that the 2D interactions modelled by our L-CPM are (due to spatial constraints) more likely to drive random phagocyte-bacteria collisions than 3D interactions would. Although this is not a problem in the specific case of the epithelium modelled here (which essentially *is* a 2D system), care must be taken when translating these findings to other tissues. Still, this 2D topology is relevant to many barrier surfaces such as the upper and lower airways, the skin, and the eyes, as well as the lining of sinuses in the liver and spleen.

Second, our simple model does not account for host responses below the epithelium (e.g., lamina propria macrophages) or the effect of the mucus layer containing normal flora above the epithelium. These are important phenomena that can be investigated in future iterations of the model with additional 2D CPM layers. Indeed, this is a significant strength of our layered CPM: it can be readily modified to accommodate more complex host-pathogen systems, without the complications arising from rendering every component of the system in 3D.

One such model extension could be aimed at examining the effect of bacterial motility over longer time scales. We here focused the dynamics of bacterial invasion and phagocytosis during the first hour after infection, but a slightly more complex model could extend this period to multiple hours. Such an extended model should take into account that in vivo, bacteria take an hour or more to transit down the intestine after oral infection, and that phagocytes are recruited in multiple steps; neutrophils extravasate from submucosal vessels and migrate through the lamina propria towards sites of bacterial infection, where they can patrol the basolateral side of the epithelium or undergo transepithelial migration to patrol the apical surface [53, 54]. The L-CPM could implement these dynamics by gradually increasing bacteria and phagocyte numbers over time.

The current model suggests that due to robust Lm motility, the earliest invasion events occur too rapidly for neutrophils to oppose them—but during persistent infection, phagocytes on the epithelium could capture motile bacteria through a "fly paper" mechanism and reduce the bacterial burden during later stages of infection. Furthermore, we often observe blood in the lumen of Lm infected mouse intestines. Since neutrophils are abundant in blood (300–500 polymorphonuclear neutrophils per µl of blood in C57/B6 mice [65]), an interesting future avenue would be to study the role of bleeding in anti-bacterial host responses.

For future work, it will also be important to develop in vitro and/or in vivo systems in which the rates of phagocytosis, epithelial attachment and invasion can be measured more precisely than we do here. With more robust estimates of phagocytosis, attachment, and invasion rates, future CPM simulations could also explore the effect of bacterial opsonization by antibody or complement on phagocytosis efficiency, bacterial motility, and invasion efficiency.

## Conclusion

In summary, we have developed an intuitive and flexible L-CPM model of host-pathogen interactions in the gut, and our results suggest that bacterial motility is a critical factor in invasion and infection outcomes. Throughout a wide range of tested parameters, our model supports the idea that Lm motility confers an invasion advantage because it allows Lm to quickly locate its preferred target cells (i.e., goblet cells). We believe that our L-CPM-based modeling approach is highly adaptable and thus could be applied to study cell dynamics in a wide range of systems, including cellular immunity to pulmonary or skin infections, host microbiota interactions, bacterial biofilm formation, antigen presentation, tumor immunology and neuroimmune cross talk.

## Supporting information

**S1 Methods. Supplemental methods.** This file contains supplemental methods and includes citations [66–71].
(PDF)

**S1 Movie. Video microscopy of Lm-37 motility in vitro.** Lm was cultured overnight at 37˚C, mounted on a glass slide and motility assessed by bright field video microscopy and automated cell tracking (Imaris). Scale bar = 50µm (Lower left); time stamp = min:sec:msec (Lower right).
(MP4)

**S2 Movie. Video microscopy of Lm-RT motility in vitro.** Lm was cultured overnight at room temperature ($\sim 25$˚C), mounted on a glass slide and motility assessed by bright field video microscopy and automated cell tracking (Imaris). Scale bar = 50µm (Lower left); time stamp = min:sec:msec (Lower right).
(MP4)

**S3 Movie. Motile Lm-RT are present in the ileum 1–1.5h after oral infection.** C57BL/6 mice were infected by gavage with $2\times10^8$ EGD-GFP Lm-RT, to test whether Lm motility persists in vivo long enough for bacteria reach the ileum. We used EGD for this experiment because it binds poorly to mouse E-cadherin and thus allows us to assess Lm motility without the confounding effects of adhering to goblet cells. Mice were sacrificed 1–1.5hpi and the ileum was imaged using 2P microscopy. Scale bar: 50 µm, time stamp: sec:msec.
(MP4)

**S4 Movie. 2P imaging of Lm-37 interaction dynamics with mouse intestinal epithelium.** Mouse ileum was explanted, infected with $1\times10^8$ Lm-37 (BacLight-Green) and epithelial interactions assessed by 2P video microscopy. Red fluorospheres were added to control for drift. Dark oval areas on the dim red background are villi. Scale bar = 50µm (Lower left); time stamp = min:sec:msec (Lower right).
(MP4)

**S5 Movie. 2P imaging of Lm-RT interaction dynamics with mouse intestinal epithelium.** Mouse ileum was explanted, infected with $1\times10^8$ Lm-RT (BacLight-Green) and epithelial interactions assessed by 2P video microscopy. Red fluorospheres were added to control for drift.

Dark oval areas on the dim red background are villi. Example Lm-RT tracks (time encoded, Imaris) are shown for Lm interacting with and binding to the epithelium. Scale bar = 50µm (Lower left); time stamp = min:sec:msec (Lower right).
(MP4)

**S6 Movie. 2P imaging of Lm-37 interaction dynamics with human intestinal epithelium.** Human ileal biopsy tissue was explanted, infected with 1x10$^8$ Lm-37 (EGD-GFP) and epithelial interactions assessed by 2P video microscopy. Red fluorospheres were added to control for drift. Dark oval areas on the dim red background are villi. Scale bar = 30µm (Lower left) and time stamp = min:sec:msec (Lower right).
(MP4)

**S7 Movie. 2P imaging of Lm-RT interaction dynamics with human intestinal epithelium.** Human ileal biopsy tissue was explanted, infected with 1x10$^8$ Lm-RT (EGD-GFP) and epithelial interactions assessed by 2P video microscopy. Red fluorospheres were added to control for drift. Dark oval areas on the dim red background are villi. Lm-RT are shown interacting with and scanning the epithelium. Scale bar = 30µm (Lower left) and time stamp = min:sec:msec (Lower right).
(MP4)

**S8 Movie. FlaA mutant Lm cultured at RT is non-motile in vitro.** 10403 (upper left panel) and flaA deletion mutant (lower left panel) Lm were cultured overnight at room temperature ($\sim$25˚C), mounted on a glass slide and motility assessed by bright field video microscopy. The image dimensions are 697x522µm and were acquired at 250msec time resolution. Zoomed movies from the boxed regions are show in the upper right (10403) and lower right (flaA deletion mutant) panels respectively.
(MP4)

**S9 Movie. FlaA mutant Lm cultured at RT is non-motile and adheres inefficiently to the epithelium of human ileal explant tissue.** Human ileal biopsy tissue was explanted, infected with 1x10$^8$ Lm-RT 10403 (top panels) and flaA deletion mutant Lm (bottom panels) and epithelial interactions assessed by 2P video microscopy. 10kD Rh-dextan was added to visualize the lumen (red). Dark oval areas on the red background are villi. Top panels, 10403 Lm-RT (green) are shown interacting with the epithelium and accumulating around goblet cell-shaped structures. Bottom panels, flaA deletion mutant Lm-RT (green) shows poor motility and fails to accumulate on the epithelium. Scale bar = 50µm (Lower left) and time stamp = min:sec:msec (Lower right).
(MP4)

**S10 Movie. A CPM simulation of bacterial motility and invasion on the epithelium.** See also Fig 4. Simulated non-motile (sLm-37) or motile (sLm-RT) bacteria, shown in blue with light-blue traces, scan the epithelium for target cells to invade (shown in a darker gray). Attached (A) and invaded (I) bacteria are shown in gray; phagocytosed (P) bacteria are invisible. Scale bar: 10 µm, timestamp in hh:mm:ss.
(MP4)

**S11 Movie. 2D motility of neutrophils on mouse intestinal epithelium in vivo.** LysM-GFP mice were infected, and neutrophils (green) on the luminal side of the epithelium (dim red) were imaged by intravital 2P microscopy. Representative individual neutrophils tracks are are shown (time encoded, Imaris). The scale bar = 20µm (Lower left) and time stamp = min:sec (Lower right).
(MP4)

**S12 Movie. Motility and invasion of non-motile (sLm-37) and motile (sLm-RT) bacteria on the epithelium in the presence of phagocytes.** The epithelium is shown in gray, with darker cells representing target cells. Scanning bacteria are shown in blue with their traces in lighter blue; attached (A) and invaded (I) bacteria are shown in gray, phagocytosed (P) bacteria are no longer shown. Phagocytes are shown as pink cells with dark protruding fronts. Scale bar: 10 μm, timestamp in hh:mm:ss. See also S10 Movie.
(MOV)

**S13 Movie. Motility and invasion of non-motile (sLm-37) and motile (sLm-RT) bacteria for varying challenge doses.** As S12 Movie, but now for varying numbers of bacteria. Scale bar: 10 μm, timestamp in hh:mm:ss. A = attached bacteria, I = invaded bacteria, P = phagocytosed bacteria.
(MOV)

**S14 Movie. Motility and invasion of non-motile (sLm-37) and motile (sLm-RT) bacteria for varying numbers of target cells.** As S12 Movie, but now for varying numbers of target cells. Scale bar: 10 μm, timestamp in hh:mm:ss. A = attached bacteria, I = invaded bacteria, P = phagocytosed bacteria.
(MOV)

**S15 Movie. Motility and invasion of non-motile (sLm-37) and motile (sLm-RT) bacteria for varying numbers of phagocytes.** As S12 Movie, but now for varying numbers of phagocytes. Scale bar: 10 μm, timestamp in hh:mm:ss. A = attached bacteria, I = invaded bacteria, P = phagocytosed bacteria.
(MP4)

**S16 Movie. Lm-RT interacting with neutrophils in vitro.** LysM-GFP mice were sacrificed, and blood was harvested and placed on a slide for imaging. $1\times10^8$ Lm-RT (BacLight-Red) were added, and Lm-neutrophil interactions imaged using 2P microscopy with frame rates of 16s (for capturing neutrophil dynamics) and 100 ms (for capturing Lm motility) respectively. Scale bars: 50 μm. Time stamps are in min:sec for the f = 16s movies and in seconds:msec for the f = 100ms movies.
(MP4)

**S17 Movie. Neutrophils on the luminal surface of the epithelium interacting with Lm-RT.** LysM-GFP mice were infected intraluminally with $1\times10^8$ Lm. Mice were sacrificed 5hpi and the ileum was explanted and rechallenged with $1\times10^8$ Lm (BacLight-Red). Lm-RT move ∼50x faster than neutrophils. Neutrophil behavior was imaged with 2P microscopy with frame rate of 24s, but bacteria move too fast to be reliably tracked in 3D with this time resolution. Therefore, we used the 100 ms frame rate recordings to track both neutrophils (despite tracking error inflating their speed) and Lm moving nearby, which controlled for potential environmental variability. Scale bars: 50 μm. Time stamps are in min:sec for the f = 24s movies and in seconds:msec for the f = 100ms movies.
(MP4)

**S1 Fig. Comparing Lm motility across populations.** A,B: Comparison of speed (A) and mean squared displacement (MSD, B) between human EGD Lm and murinized Lm$^{Mu}$. C: MSD curve of the three Listeria populations (EGD Lm-37, Lm-RT, Lm-RT-m), fitted by Fürth's equation (red line). The red dashed line indicates the persistence time as determined from the fit. Curves were fit on Δt up to 5s; for longer Δt, fast cells tend to leave the imaging window and the MSD becomes biased (see Methods for details). Lm-37 did not move and could not be fitted by

Fürth's equation. D: Autocovariance curve of the populations as fitted by an exponential decay: $f(x) = f_0{}^*\exp(-x/P)$, red line. This yields a slightly higher estimate of the persistence time P (vertical dashed red lines), but still in the same order of magnitude as those in panel C. E,F: To estimate uncertainty in motility parameters estimated from the MSD (motility coefficient M and persistence time P), tracks were resampled from the original populations with replacement N = 1000 times, to obtain N "bootstrapped" datasets of equal size as the original. Resampled datasets were then fitted with Fürth's equation as shown in panel A to obtain N estimates of M and P. G: To assess how long Lm-RT stay motile after being placed at 37˚C, Lm-InlA was first grown at RT with shaking (200rpm) to OD600 around 1.0, and then switched to a 37˚C incubator (shaking at 200rpm) to assess motility after 2–3 hours. Samples were diluted 1:10 with BHI and plated on a non-charged slide (Globe Scientific Cat No 1324W); a 2D time lapse video was recorded for 15 seconds with 250ms time and 100ms exposure using Olympus IX51 inverted microscope with 20X objective and phase dichroic filter. Cells were were tracked in Imaris 9.3 and % motile cells were calculated using 4 μm track displacement length filter.
(TIF)

**S2 Fig. Removing non-motile cells from the Lm-RT population to create a more representative motile data set.** To remove artefacts of non-motile cells sticking to the glass slide, bacteria separate tracks were separated into "motile" vs "static" tracks. Briefly, tracks were considered motile whenever the track coordinates were described better by two Gaussian distributions (splitting the track in two parts) than by a single Gaussian. If a single Gaussian distribution was a reasonably good fit for the observed coordinate, the tracks were classified as static. See S1 Methods for details. A,B: static (orange) and motile (blue) cells of Lm-37 and Lm-RT, shown in the speed distribution. Whereas static tracks tend to have low speeds, there are also some static tracks with relatively high speeds (mostly when the track contains a single motile step while the cell otherwise does not move). Blue tracks represent the "Lm-RT-m" population. C,D: tracks of Lm-37 and Lm-RT, showing that the filter indeed reasonably removes non-motile cells. Zoomed inset: 50 x 50 μm.
(TIF)

**S3 Fig. Motile GFP-Lm-RT arrive in the ileum rapidly after oral infection.** C57BL/6 mice were infected by gavage with 2x10[8] EGD-GFP Lm, which bind poorly to mouse E-cadherin and thus allows motility to be assessed in the absence of goblet cell recognition and invasion. Mice were sacrificed 1–1.5hpi and the ileum was imaged using 2P microscopy. A,B: show motility behaviors in two different regions; in A, Lm was imaged near the villi (white arrows) as well as in the mucus and fluid phases, with tracks exhibiting both short and long persistence. The zoomed insets show time-encoded tracks of motile Lm over 12 seconds. Examples of epithelial scanning can be observed in the zoomed panel from A. Scale bar: 50μm.
(TIF)

**S4 Fig. FlaA-mutant Lm cultured at RT behave similar to WT Lm-37.** Human ileal biopsy tissue was explanted and infected with 1x10[8] Lm-RT 10403 or flaA deletion mutant Lm. Epithelial interactions were assessed by 2P video microscopy using 10kD Rh-dextan to visualize the lumen (red). A,B: Example images for 10403 Lm (A) and the flaA mutant strain (B). Zoomed insets show examples of Lm (yellow arrows) at the epithelial surface (white arrows). Dark oval areas on the red background are villi. Scale bar: 50 μm. C: Quantification of Lm (green voxels) overlapping the epithelial surface in both strains, showing that at RT, flaA mutant Lm is deficient in reaching the epithelial surface (similar to Lm-37; see also Figs 2H, 3J and 3K). 10403 Lm overlapped with the epithelial surface about 16 times more than flaA mutant Lm did (95% CI: [8.3–33]). Each point represents an image, from a total of 2 mice per

condition. D: 10403 Lm (green, yellow arrows) co-localized near large epithelial cells that were both Rh-dextran and Cytokeratin-18 staining, consistent with goblet cells. E: Lm flaA deletion mutant colocalization was comparably less (dim green, yellow arrow) and Lm often aggregated in clumps in the mucus layer (bright green, yellow arrow). In D,E, Scale bar: 100 μm.
(TIF)

**S5 Fig. Matching motility of simulated Lm to in vitro data.** Average bacterial motility (of 50 simulated bacteria) in the model (sLm-RT) closely matches in vitro motility (Lm-RT) in both speed and directionality, as shown by: A: the distribution of cell speeds, B: the mean squared displacement (MSD) over different time intervals Δt and C: the (normalized) autocovariance of movement "step" vectors with time Δt between them (the longer it takes for this curve to drop to zero, the larger the persistence time of the cells). See Methods for details.
(TIF)

**S6 Fig. Target cell infection efficiency as a function of sLm speed.** sLm speed was changed by varying the $v_{rel}$ parameter in the model. This parameter controls how many "steps" of the bacterial model occur each second; high values increase bacterial speed whereas a value of zero means that bacteria are completely static (the default value used throughout the paper is 150 to simulate sLm-RT and 1 to simulate sLm-37). Target cell infection efficiency is measured as A: the % of sLm that have invaded after 60 min for all tested speeds (top) or for sLm speeds up to 10 steps/s (bottom; corresponding to the gray region in the upper plot); and B: likewise, but now for the % of target cells that has been invaded by sLm. Results are shown as mean ± SE for 20 independent simulations. Horizontal dashed lines in A represent the % of the surface area covered with target cells.
(TIF)

**S7 Fig. Target infection efficiency as a function of the attachment rate $k_{attach}$ of sLm to target cells.** When $k_{attach}$ is high, scanning the epithelium will immediately attach to any target cell they encounter; when it is low, sLm are more likely to move past target cells instead of attaching to them (default value used in the paper: 0.051 s$^{-1}$). Target infection efficiency is measured as A: the % of sLm that has invaded after 60 min for all tested speeds (top) or for sLm speeds up to 10 steps/s (bottom; corresponding to the gray region in the upper plot); and B: likewise, but now for the % of target cells that has been invaded by sLm. Results are shown as mea ± SE for 20 independent simulations.
(TIF)

**S8 Fig. Substantial numbers of neutrophils have transmigrated to the apical side of the epithelium at 5hpi.** A,B: LysM-GFP mice were treated intraluminally with either vehicle (sham) or 2x10$^8$ Lm. Mice were sacrificed, the ileum explanted, and 3D images collected from the luminal side to assess neutrophil (LysM-GFP) recruitment to the surface of the epithelium. Scale bar: 50 μm. C: GFP cell numbers on the epithelium (examples shown by yellow arrows), were enumerated using the spot function in Imaris and compared using a two-tailed Mann-Whitney non-parametric test. Data are from 12 (sham) and 16 (Lm-RT) images from 3 independent mice. D: Upon rechallenge, transmigrated neutrophils interact with incoming Lm at the epithelium. LysM-GFP mice were treated intraluminally with 1x10$^8$ Lm. Mice were sacrificed, the ileum explanted, rechallenged with 1x10$^8$ Lm (BacLight-Red labelled) and 2P imaged alongside neutrophils (LysM-GFP)at the epithelium. Scale bar: 20 μm. E: Example of Lm (red, red arrows) phagocytosis by LysM-GFP neutrophils (green). Left: overview with three examples of phagocytosis (scale bar: 25 μm), including a -90˚ view. Right: zoomed view for the highlighted example at two different time points (scale bar: 20μm).
(TIF)

**S9 Fig. 2D motility of neutrophils on mouse intestinal epithelium in vivo.** Mice were anesthetized, the ileum glued to a plastic support and carefully dissected to expose the luminal surface for 2P microscopy. A: Blood vessels were labeled with 655nm Q-dots and images of the epithelium acquired and analyzed to estimate epithelial cell dimensions and structure for the L-CPM. B: LysM-GFP mice were imaged with time-lapse 2P microscopy to assess neutrophil (green) migration dynamics on the surface of the epithelium (red). Multidimensional datasets were rendered, and cells tracked in Imaris. Tracks are time encoded. Scale bar = 20μm. Time stamp is min:sec. Neutrophil motility parameters were calculated using celltrackR/MotilityLab (2Ptrack.net) and used as the basis for phagocyte motility in the L-CPM model.
(TIF)

**S10 Fig. Matching simulated phagocyte motility to in vivo neutrophil motility.** Phagocyte motility in the model (average of 40 simulated cells) closely matches in vivo motility of neutrophils crawling between epithelium and coverglass. Motility matches in both speed and directionality, as shown by: **A**, the distribution of cell speeds, **B**, the mean squared displacement (MSD) over different time intervals Δt and **C**, the (normalized) autocovariance of movement "step" vectors with time Δt between them (the longer it takes for this curve to drop to zero, the larger the persistence time of the cells). See Methods for details.
(TIF)

**S11 Fig. Dependency of immunological outcomes on numbers of bacteria, target cells, and phagocytes.** Immunological outcomes were assessed after varying (A-D): the number of sLm challenged (with A: % sLm invaded, B: % sLm phagocytosed, C: % of target cells invaded by sLm after 60 min, and D: the data from panels A-B compared in one plot for motile sLm-RT). While challenge dose does not affect the % of sLm invading or phagocytosed for motile sLm-RT (blue), it does for low doses of the non-motile sLm-37 (gray), which are more likely to diffuse on or near target cells before being phagocytosed. Dependency of immunological outcomes was similarly tested for E-H: the number of "target" cells (i.e. goblets that can be invaded; the total number of epithelial cells is 289), and I-L: the number of phagocytes. Results are shown as mean ± standard deviation (SD) for 20 independent simulations for motile (sLm-RT, blue) and non-motile bacteria (sLm-37, gray). Vertical dotted lines indicate the baseline of 100 sLm, 20 target cells, and 14 phagocytes used in the rest of the paper.
(TIF)

**S12 Fig. Dependency of immunological outcomes on invasion and phagocytosis kinetics.** Immunological outcomes were assessed after varying (A-D): the rate by which sLm attach to target cells they encounter ($k_{attach}$) or (E-G): the rate by which sLm are phagocytosed by encountered phagocytes ($k_\varphi$). Plots represent the following: A,E: % sLm invaded, B,F: % sLm phagocytosed, C,G: % of target cells invaded after 60 min, and D,H: the data from panels A-B compared in one plot for motile sLm-RT. Results are shown as mean ± SD for 20 independent simulations for motile sLm-RT (blue) and non-motile sLm-37 (gray). Vertical dotted lines indicate the default values of $k_{attach} = k_\varphi = 0.051$ s$^{-1}$ used in the rest of the paper.
(TIF)

**S13 Fig. Dependency of immunological outcomes on sLm and phagocyte speed.** Immunological outcomes were assessed after varying A-C: the relative sLm speed in steps/s (see also S6 Fig), and D-F: phagocyte speed. Plots show A: % invaded sLm, B: % phagocytosed sLm and C: % target cells invaded after 5 or 60 min, with horizontal lines in A-B indicating the percentage of the surface area covered by target cells or phagocytes, respectively. The bottom plots in A-C are zoomed in on lower relative speeds, indicated by the gray shaded region in the top-row

panels. D-F show only the 60-minute curve, but now for motile phagocytes (modelled with default parameters, pink) compared to non-motile phagocytes (modelled with $\lambda_{act} = max_{act} = 0$, gray). Phagocyte motility affects outcomes only in the (very) low range of sLm motility, where bacteria are static enough that phagocytes can actually "hunt" them. All lines represent mean ± SE of 20 independent simulations.

(TIF)

## Acknowledgments

We would like to thank Johannes Textor for many helpful discussions and his expert advice. We would like to thank Charles Parkos, Ronen Sumagin, Vero Azcutia Criado and Mathias Kelm for the E-cadherin-CFP mice. We would like to credit the WashU In Vivo Imaging Core for 2P microscopy, Wandy L. Beatty and the Molecular Microbiology Imaging Facility for electron microscopy, and Lihua Yang for technical help with immunofluorescence microscopy. We would like to thank Lisa Gorsky for sending us the flaA deletion mutant Lm and parental strains. Finally, we thank Jérémy Postat for helpful comments and Gijs Schröder for testing the usability of the code in the online repository.

## Author Contributions

**Conceptualization:** Inge M. N. Wortel, Seonyoung Kim, Mark J. Miller.

**Data curation:** Inge M. N. Wortel, Seonyoung Kim, Enid C. Ibarra, Mark J. Miller.

**Formal analysis:** Inge M. N. Wortel, Seonyoung Kim, Annie Y. Liu, Enid C. Ibarra, Mark J. Miller.

**Funding acquisition:** Mark J. Miller.

**Investigation:** Inge M. N. Wortel, Seonyoung Kim, Mark J. Miller.

**Methodology:** Inge M. N. Wortel, Seonyoung Kim, Annie Y. Liu, Enid C. Ibarra, Mark J. Miller.

**Project administration:** Seonyoung Kim, Mark J. Miller.

**Resources:** Mark J. Miller.

**Software:** Inge M. N. Wortel.

**Supervision:** Inge M. N. Wortel, Seonyoung Kim, Mark J. Miller.

**Visualization:** Inge M. N. Wortel.

**Writing – original draft:** Inge M. N. Wortel, Seonyoung Kim, Mark J. Miller.

**Writing – review & editing:** Inge M. N. Wortel, Mark J. Miller.

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
