## [Decision Letter · Decision Letter 0]

11 Sep 2022

Dear Dr. Miller,

Thank you very much for submitting your manuscript "Listeria motility increases the efficiency of goblet cell invasion during intestinal infection" for consideration at PLOS Pathogens. As with all papers reviewed by the journal, your manuscript was reviewed by members of the editorial board and by several independent reviewers. In light of the reviews (below this email), we would like to invite the resubmission of a significantly-revised version that takes into account the reviewers' comments.

We cannot make any decision about publication until we have seen the revised manuscript and your response to the reviewers' comments. Your revised manuscript is also likely to be sent to reviewers for further evaluation.

Sincerely,

Leigh Knodler

Guest Editor

PLOS Pathogens

Raphael Valdivia

Section Editor

PLOS Pathogens

Kasturi Haldar

Editor-in-Chief

PLOS Pathogens

orcid.org/0000-0001-5065-158X

Michael Malim

Editor-in-Chief

PLOS Pathogens

orcid.org/0000-0002-7699-2064

Reviewer's Responses to Questions

**Part I - Summary**

Reviewer #1: Several gut bacterial pathogens use flagella to propel themselves in the intestinal lumen, approach the epithelium, and promote invasion into this barrier. However, for certain pathogens, including Yersinia and Listeria strains, the role of flagella in host infection has been less obvious. This because flagellar expression appears to be stimulated at conditions mimicking the external environment (<<37 ˚C), but suppressed upon transition to host temperature (37 ˚C).

In this manuscript, Wortel&Kim et al investigate Listeria monocytogenes (Lm) motility at external (~23˚C; “RT”) versus host (37 ˚C) temperature. They combine analysis of motility parameters in pure bacterial cultures with 2-photon imaging of Lm inoculated onto murine and human intestinal explants. This allows the authors to assess how motile and non-motile Lm scout the epithelium for Goblet cells, previously established as favorable targets for Lm mucosal invasion. Finally, a computational model, partly based on experimentally determined parameters, is devised to predict how Lm motility impacts epithelial target search and interactions with phagocytes at this barrier.

This study has several merits, particularly it: 1) tackles a significant question of how the seemingly “inverse temperature-regulation” of Lm motility may make sense; 2) assesses Lm motility and its consequences atop intact intestinal tissue, and; 3) provides steps towards a comprehensive computational model that predicts infection outcome in the gut mucosa.

Both the microscopy and the numerical analyses appear of high quality. Still, there are a couple of considerable leaps in the reasoning that lack grounding in experimental data. These topics need to be addressed to make the study fully coherent and credible.

Reviewer #2: Wortel et al sought to address whether bacterial flagellar motility, which in Listeria is known to be repressed by growth at 37 degrees, could help Listeria traverse the intestinal epithelium to locate and invade goblet cells based on the hypothesis that contaminated food would likely be stored in the refrigerator or freezer and thus could be motile in the GI tract. They have used two-photon microscopy of intestinal explants from mouse and human tissue, cocultures with neutrophils and computer modeling to address this question. Overall, the manuscript is interesting and innovative and they use new tools which may shed light on questions that are difficult to address in traditional mouse models and tissue culture. I have some suggestions and concerns, however, that would strengthen the conclusions of the paper.

Reviewer #3: The authors examine the role of motility in driving early infection of the intestine by Listeria monocytogenes. They combine live cell imaging of an explant tissue model (both mouse and human) with a cellular pots model to examine early interactions of Listeria with both target cells (goblet cells) and phagocytes. Their findings suggest that bacterial motility promotes infection of the intestine by promoting interactions with target host cells. Furthermore, their findings provide insight into how phagocytes can protect the intestine by serving as “fly paper” rather than chasing invading bacteria. This is an interesting paper and I think will promote important discussions about host-pathogen interactions in the gut. I have mostly minor comments for the authors to consider.

**Part II – Major Issues: Key Experiments Required for Acceptance**

Reviewer #1: 1. In the first part of the study (Fig 1-3) the authors make the implicit assumption that only motility is relevant to consider when comparing how Lm-RT vs Lm-37˚C scout the epithelium and invade goblet cells. However, temperature shift will alter many aspects of virulence and it is known that the Lm master virulence regulator, PrfA, is under temperature control. To substantiate their claims, the authors should ensure that a flagellar mutant of Lm grown at RT behaves as the wild-type Lm grown at 37 ˚C with respect to invasion of Goblet cells (in connection to results presented in Fig 2-3).

2. For the findings to be physiologically relevant, Lm-RT need to retain flagellar motility upon arrival in the intestinal lumen after a per-oral infection. Indeed, the authors show that Lm-RT switched to 37 ˚C in liquid broth retain motility for up to 3h (Fig S1), which is a good start. However, it should be straight-forward to conduct a per-oral infection in mice with labelled Lm-RT and assess to what extent motile Lm can be observed in gut luminal content at different time-points post-infection. This appears critical to support the main conclusion of the manuscript.

3. The last section of the study, which simulates the presence of phagocytes atop the epithelial lining, includes a speculative leap. The authors state that it is unknown from what time-point of infection, and in what numbers, phagocytes arrive at the apical side of the epithelium. Knowledge of these basic parameters is key for the relevance of the simulations in Figure 5. Again, this could be addressed experimentally in a straight forward manner. In similar per-oral Lm infections as in my comment 2, fixed intestinal tissue could be cross-sectioned, stained for phagocyte and epithelial cell markers, and the number of phagocytes present at the apical side of the epithelium quantified across time-points. Based on experience from other gut infection models, I however predict that during the first hours of infection, where Lm may remain motile, close to zero transmigrated phagocytes will be observed. This is typically a later phenomenon coupled to the inflammatory response. Nevertheless, the temporal relationship between Lm motility in the lumen (comment 2) and arrival of phagocytes atop the epithelium needs to be sorted out.

Reviewer #2: Major points:

The authors rely on growth at 37 or 23 of one strain of Listeria and refer to studies in the literature which use mutants in bacterial motility with conflicting results, however they have generated a new model system in which they could and should test a bacterial chemotaxis or motility mutant to support their conclusions that bacteria are actively migrating towards goblet cells.

1) In particular, they take videos with the bacteria migrating to “goblet cell” like structures but in the same video there are other green bacteria that have not been tracked or that do not move towards goblet cells. How were the bacteria with tracks selected for quantification? Are these data quantified? I mainly saw that they quantified the change in CFUs, which I think is extrapolated from imaging or conducted through separate experiments with mouse infection.

2) If the authors want to say that the bacteria are moving towards goblet cells from the videos, the authors should confirm that these cells are indeed what they say that they are using a goblet cell marker such as MUC5A or another mucin. If they cannot do so with two-photon they could use fixed images from their explants. Similarly, using Rh-dextran as a marker of goblet cells later on would not differentiate between goblet cells and phagocytic cells (from the explant) that could also take up the dye. I assume that both mouse explants and human tissue explants could have associated phagocytic cells but please correct me if I am wrong.

3) For the human explants the authors could treat the tissue with gentamicin to assess internalized bacterial cells in addition to extrapolating CFUs based on GFP pixel intensity. They could also measure CFUs on the other side of the tissue if the bacteria can transcytose across the tissue layer. This has been done with mouse intestinal tissue before.

4) The EGD strain of Listeria has a published mutation in PrfA, thus the virulence factors are constitutively produced in this strain. How is motility affected? Does PrfA negatively regulate motility? Could an abundance of virulence factors (InlA in particular) affect the conclusions of the study?

5) How long do flagella persist when switched to 37 across the various strains of bacteria? The authors mention that a certain percentage of bacteria still express flagella but if they collected CFUs shed from animals following digestion would the bacteria still be motile?

6) While some neutrophils extravasate – adding a neutrophil layer on the apical side of the gut seems to confuse the message as the number of neutrophils patrolling would probably not reach that of the lumenal side of the epithelial layer – why did the authors not address the question of motility in co-culture in vitro of neutrophils and motile or non-motile bacteria without the epithelial layer?

These conclusions of the model that bacterial motility facilitates phagocytosis are testable but I’m not sure that their model tests it under realistic conditions. It is hard to extrapolate this conclusion if phagocytes would not normally patrol the apical side of the layer. There is an entire microbiome that would be on that layer of cells, so maybe it would be more relevant to test bacterial density and its affect on Lm chemotaxing towards goblet cells rather than neutrophils which would not really be there in the density simulated.

There is also a very established literature of phagocytes chemotaxing towards bacteria and in vitro data showing that it occurs so overturning this literature by saying that “simulating bacterial and phagocyte dynamics to scale, our CPM clearly shows that neutrophils cannot possibly “hunt” motile bacteria – which move at their own speed” is difficult to believe when the neutrophils would not be hunting the motile bacteria, unless the motile bacteria retain their flagella on the lumenal side of the epithelium. The authors could easily test this finding by putting motile bacteria with neutrophils in vitro to see if the neutrophils are fast enough to catch the bacteria.

If I have not understood that aspect of the model – maybe more explanation is needed for a microbiology audience.

Reviewer #3: -Figure 2A-F. The images are very hard to interpret. A schematic showing orientation of the tissue and colour code for labels would help guide the reader. What is blue? The panel images indicate DAPI but the figure legend indicates autofluorescence.

-Figure 2H. This data seems like it should be panel A. Also, were other tissues collected for CFU counts?

-Figure 3I. The green rings of Lm observed are fascinating. The authors presume that Lm is binding to E-cadherin on goblet cells. However, Figure 3C shows that E-cadherin does not have a strong signal around the periphery of what are presumed goblet cells. Is this just a matter of the image chosen? Or are the rings created by Lm binding to themselves as well as the host cell?

**Part III – Minor Issues: Editorial and Data Presentation Modifications**

Reviewer #1: 4. Abstract, Author summary, beginning of Discussion et.c.: Here, the authors make statements like “little is known regarding the effect of bacterial motility on invasion…”. This may be true for Lm, but not for gut bacterial infection in general. In particular for Salmonella, a string of published papers has detailed how flagellar motility enables the pathogen to find mucus gaps, efficiently reach the epithelial surface, and scout it by near-surface swimming to locate ideal invasion sites. The authors are advised to rephrase these passages to better reflect the current knowledge front.

5. Page 3, line 13: “One example is the forming protein”. The word “pore” is missing.

6. Page 3, line 31 and elsewhere: The authors talk about “immunological outcomes” at many places in the text. However, this manuscript completely lacks data on immunological outcomes (i.e. changes in histopathology, induction of cytokine production, immune cell recruitment into the mucosa, antibody levels in blood or similar). Hence, another wording would be more appropriate.

7. Page 4, line 64-65: The % motile Lm under these different conditions is an important metric, but is hidden in Fig S1. Please clarify.

8. Figure 2-3 legends and results text: The reader has to go to the Methods to figure out which gut segment has been used. Please clarify.

9. Legends to Figure S6 and Supplementary Movies S1-S6, S8: These lack information on scale bars, color schemes, time stamps et.c. Please amend.

Reviewer #2: (No Response)

Reviewer #3: -line 83, “Sections of B6 mouse intestine were harvested, glued to plastic…”. A schematic would help guide the reader as to how tissues were processed and infections performed.

-line 264 …”Lm-RT (both EDG and 10403S strains)”… should be EGD

PLOS authors have the option to publish the peer review history of their article (what does this mean?). If published, this will include your full peer review and any attached files.

Reviewer #1: No

Reviewer #2: No

Reviewer #3: No
---

## [Editor Report · Decision Letter 1]

28 Nov 2022

Dear Dr. Miller,

We are pleased to inform you that your manuscript 'Listeria motility increases the efficiency of epithelial invasion during intestinal infection' has been provisionally accepted for publication in PLOS Pathogens.

Best regards,

Leigh Knodler

Guest Editor

PLOS Pathogens

Raphael Valdivia

Section Editor

PLOS Pathogens

Kasturi Haldar

Editor-in-Chief

PLOS Pathogens

orcid.org/0000-0001-5065-158X

Michael Malim

Editor-in-Chief

PLOS Pathogens

orcid.org/0000-0002-7699-2064
---

## [Editor Report · Acceptance letter]

21 Dec 2022

Dear Dr. Miller,

We are delighted to inform you that your manuscript, "Listeria motility increases the efficiency of epithelial invasion during intestinal infection," has been formally accepted for publication in PLOS Pathogens.

Best regards,

Kasturi Haldar

Editor-in-Chief

PLOS Pathogens

orcid.org/0000-0001-5065-158X

Michael Malim

Editor-in-Chief

PLOS Pathogens

orcid.org/0000-0002-7699-2064